# DISTRIBUTION COMPRESSION IN NEAR-LINEAR TIME

**Abhishek Shetty**[1]**, Raaz Dwivedi**[2]**, Lester Mackey**[3]

[1] Department of EECS, UC Berkeley
[2] Department of Computer Science, Harvard University and Department of EECS, MIT
[3] Microsoft Research New England
SHETTY@BERKELEY.EDU, RAAZ@MIT.EDU, LMACKEY@MICROSOFT.COM

## ABSTRACT

In distribution compression, one aims to accurately summarize a probability distribution $\mathbb{P}$ using a small number of representative points. Near-optimal thinning procedures achieve this goal by sampling $n$ points from a Markov chain and identifying $\sqrt{n}$ points with $\widetilde{\mathcal{O}}(1/\sqrt{n})$ discrepancy to $\mathbb{P}$. Unfortunately, these algorithms suffer from quadratic or super-quadratic runtime in the sample size $n$. To address this deficiency, we introduce Compress++, a simple meta-procedure for speeding up any thinning algorithm while suffering at most a factor of $4$ in error. When combined with the quadratic-time kernel halving and kernel thinning algorithms of Dwivedi and Mackey (2021), Compress++ delivers $\sqrt{n}$ points with $\mathcal{O}(\sqrt{\log n/n})$ integration error and better-than-Monte-Carlo maximum mean discrepancy in $\mathcal{O}(n \log^3 n)$ time and $\mathcal{O}(\sqrt{n} \log^2 n)$ space. Moreover, Compress++ enjoys the same near-linear runtime given any quadratic-time input and reduces the runtime of super-quadratic algorithms by a square-root factor. In our benchmarks with high-dimensional Monte Carlo samples and Markov chains targeting challenging differential equation posteriors, Compress++ matches or nearly matches the accuracy of its input algorithm in orders of magnitude less time.

## 1 INTRODUCTION

Distribution compression—constructing a concise summary of a probability distribution—is at the heart of many learning and inference tasks. For example, in Monte Carlo integration and Bayesian inference, $n$ representative points are sampled i.i.d. or from a Markov chain to approximate expectations and quantify uncertainty under an intractable (posterior) distribution (Robert & Casella, 1999). However, these standard sampling strategies are not especially concise. For instance, the Monte Carlo estimate $\mathbb{P}_{\text{in}} f \triangleq \frac{1}{n} \sum_{i=1}^{n} f(x_i)$ of an unknown expectation $\mathbb{P}f \triangleq \mathbb{E}_{X \sim \mathbb{P}}[f(X)]$ based on $n$ i.i.d. points has $\Theta(n^{-\frac{1}{2}})$ integration error $|\mathbb{P}f - \mathbb{P}_{\text{in}}f|$, requiring 10000 points for 1% relative error and $10^6$ points for 0.1% error. Such bloated sample representations preclude downstream applications with critically expensive function evaluations like computational cardiology, where a 1000-CPU-hour tissue or organ simulation is required for each sample point (Niederer et al., 2011; Augustin et al., 2016; Strocchi et al., 2020).

To restore the feasibility of such critically expensive tasks, it is common to thin down the initial point sequence to produce a much smaller coreset. The standard thinning approach, select every $t$-th point (Owen, 2017), while being simple often leads to an substantial increase in error: e.g., standard thinning $n$ points from a fast-mixing Markov chain yields $\Omega(n^{-\frac{1}{4}})$ error when $n^{\frac{1}{2}}$ points are returned. Recently, Dwivedi & Mackey (2021) introduced a more effective alternative, *kernel thinning* (KT), that provides near optimal $\widetilde{\mathcal{O}}_d(n^{-\frac{1}{2}})$ error when compressing $n$ points in $\mathbb{R}^d$ down to size $n^{\frac{1}{2}}$. While practical for moderate sample sizes, the runtime of this algorithm scales quadratically with the input size $n$, making its execution prohibitive for very large $n$. Our goal is to significantly improve the runtime of such compression algorithms while providing comparable error guarantees.

**Problem setup** Given a sequence $\mathcal{S}_{\text{in}}$ of $n$ input points summarizing a target distribution $\mathbb{P}$, our aim is to identify a high quality coreset $\mathcal{S}_{\text{out}}$ of size $\sqrt{n}$ in time nearly linear in $n$. We measure coreset quality via its integration error $|\mathbb{P}f - \mathbb{P}_{\mathcal{S}_{\text{out}}} f| \triangleq |\mathbb{P}f - \frac{1}{|\mathcal{S}_{\text{out}}|} \sum_{x \in \mathcal{S}_{\text{out}}} f(x)|$ for functions

$f$ in the reproducing kernel Hilbert space (RKHS) $\mathcal{H}_{\mathbf{k}}$ induced by a given kernel $\mathbf{k}$ (Berlinet & Thomas-Agnan, 2011). We consider both single function error and kernel *maximum mean discrepancy* (MMD, Gretton et al., 2012), the worst-case integration error over the unit RKHS norm ball:

$$\mathrm{MMD}_{\mathbf{k}}(\mathbb{P}, \mathbb{P}_{\mathcal{S}_{\mathrm{out}}}) \triangleq \sup_{\|f\|_{\mathbf{k}} \leq 1} |\mathbb{P}f - \mathbb{P}_{\mathcal{S}_{\mathrm{out}}} f|. \qquad (1)$$

**Our contributions** We introduce a new simple meta procedure—COMPRESS++—that significantly speeds up a generic thinning algorithm while simultaneously inheriting the error guarantees of its input up to a factor of 4. A direct application of COMPRESS++ to KT improves its quadratic $\Theta(n^2)$ runtime to near linear $\mathcal{O}(n \log^3 n)$ time while preserving its error guarantees. Since the $\widetilde{\mathcal{O}}_d(n^{-\frac{1}{2}})$ KT MMD guarantees of Dwivedi & Mackey (2021) match the $\Omega(n^{-\frac{1}{2}})$ minimax lower bounds of Tolstikhin et al. (2017); Phillips & Tai (2020) up to factors of $\sqrt{\log n}$ and constants depending on $d$, KT-COMPRESS++ also provides near-optimal MMD compression for a wide range of $\mathbf{k}$ and $\mathbb{P}$. Moreover, the practical gains from applying COMPRESS++ are substantial: KT thins $65,000$ points in 10 dimensions in 20m, while KT-COMPRESS++ needs only 1.5m; KT takes more than a day to thin $250,000$ points in 100 dimensions, while KT-COMPRESS++ takes less than 1hr (a $32\times$ speed-up). For larger $n$, the speed-ups are even greater due to the order $\frac{n}{\log^3 n}$ reduction in runtime.

COMPRESS++ can also be directly combined with any thinning algorithm, even those that have suboptimal guarantees but often perform well in practice, like kernel herding (Chen et al., 2010), MMD-critic (Kim et al., 2016), and Stein thinning (Riabiz et al., 2020a), all of which run in $\Omega(n^2)$ time. As a demonstration, we combine COMPRESS++ with the popular kernel herding algorithm and observe $45\times$ speed-ups when compressing $250,000$ input points. In all of our experiments, COMPRESS++ leads to minimal loss in accuracy and, surprisingly, even improves upon herding accuracy for lower-dimensional problems.

Most related to our work are the merge-reduce algorithms of Matousek (1995); Chazelle & Matousek (1996); Phillips (2008) which also speed up input thinning algorithms while controlling approximation error. In our setting, merge-reduce runs in time $\Omega(n^{1.5})$ given an $n^2$-time input and in time $\Omega(n^{(\tau+1)/2})$ for slower $n^\tau$-time inputs (see, e.g., Phillips, 2008, Thm. 3.1). In contrast, COMPRESS++ runs in near-linear $\mathcal{O}(n \log^3 n)$ time for any $n^2$-time input and in $\mathcal{O}(n^{\tau/2} \log^\tau n)$ time for slower $n^\tau$-time inputs. After providing formal definitions in Sec. 2, we introduce and analyze COMPRESS++ and its primary subroutine COMPRESS in Secs. 3 and 4, demonstrate the empirical benefits of COMPRESS++ in Sec. 5, and present conclusions and opportunities for future work in Sec. 6.

**Notation** We let $\mathbb{P}_{\mathcal{S}}$ denote the empirical distribution of $\mathcal{S}$. For the output coreset $\mathcal{S}_{\mathrm{ALG}}$ of an algorithm ALG with input coreset $\mathcal{S}_{\mathrm{in}}$, we use the simpler notation $\mathbb{P}_{\mathrm{ALG}} \triangleq \mathbb{P}_{\mathcal{S}_{\mathrm{ALG}}}$ and $\mathbb{P}_{\mathrm{in}} \triangleq \mathbb{P}_{\mathcal{S}_{\mathrm{in}}}$. We extend our MMD definition to point sequences $(\mathcal{S}_1, \mathcal{S}_2)$ via $\mathrm{MMD}_{\mathbf{k}}(\mathcal{S}_1, \mathcal{S}_2) \triangleq \mathrm{MMD}_{\mathbf{k}}(\mathbb{P}_{\mathcal{S}_1}, \mathbb{P}_{\mathcal{S}_2})$ and $\mathrm{MMD}_{\mathbf{k}}(\mathbb{P}, \mathcal{S}_1) \triangleq \mathrm{MMD}_{\mathbf{k}}(\mathbb{P}, \mathbb{P}_{\mathcal{S}_1})$. We use $a \precsim b$ to mean $a = \mathcal{O}(b)$, $a \succsim b$ to mean $a = \Omega(b)$, $a = \Theta(b)$ to mean both $a = \mathcal{O}(b)$ and $a = \Omega(b)$, and log to denote the natural logarithm.

## 2   THINNING AND HALVING ALGORITHMS

We begin by defining the thinning and halving algorithms that our meta-procedures take as input.

**Definition 1 (Thinning and halving algorithms)** *A thinning algorithm* ALG *takes as input a point sequence* $\mathcal{S}_{\mathrm{in}}$ *of length* $n$ *and returns a (possibly random) point sequence* $\mathcal{S}_{\mathrm{ALG}}$ *of length* $n_{\mathrm{out}}$. *We say* ALG *is* $\alpha_n$-*thinning if* $n_{\mathrm{out}} = \lfloor n/\alpha_n \rfloor$ *and* root-thinning *if* $\alpha_n = \sqrt{n}$. *Moreover, we call* ALG *a* halving algorithm *if* $\mathcal{S}_{\mathrm{ALG}}$ *always contains exactly* $\lfloor \frac{n}{2} \rfloor$ *of the input points.*

Many thinning algorithms offer high-probability bounds on the integration error $|\mathbb{P}_{\mathcal{S}_{\mathrm{in}}} f - \mathbb{P}_{\mathcal{S}_{\mathrm{ALG}}} f|$. We capture such bounds abstractly using the following definition of a sub-Gaussian thinning

**Definition 2 (Sub-Gaussian thinning algorithm)** *For a function* $f$, *we call a thinning algorithm* ALG $f$-*sub-Gaussian with parameter* $\nu$ *and write* ALG $\in \mathcal{G}^f(\nu)$ *if*

$$\mathbb{E}[\exp(\lambda(\mathbb{P}_{\mathcal{S}_{\mathrm{in}}} f - \mathbb{P}_{\mathcal{S}_{\mathrm{ALG}}} f)) \mid \mathcal{S}_{\mathrm{in}}] \leq \exp\left(\frac{\lambda^2 \nu^2(n)}{2}\right) \quad \textit{for all} \quad \lambda \in \mathbb{R}.$$

Def. 2 is equivalent to a sub-Gaussian tail bound for the integration error, and, by Boucheron et al. (2013, Section 2.3), if ALG $\in \mathcal{G}^f(\nu)$ then $\mathbb{E}[\mathbb{P}_{\mathcal{S}_{\mathrm{ALG}}} f \mid \mathcal{S}_{\mathrm{in}}] = \mathbb{P}_{\mathcal{S}_{\mathrm{in}}} f$ and, for all $\delta \in (0, 1)$,

$$|\mathbb{P}_{\mathcal{S}_{\mathrm{in}}} f - \mathbb{P}_{\mathcal{S}_{\mathrm{ALG}}} f| \leq \nu(n) \sqrt{2 \log(\tfrac{2}{\delta})}, \text{ with probability at least } 1 - \delta \text{ given } \mathcal{S}_{\mathrm{in}}.$$

Hence the integration error of ALG is dominated by the sub-Gaussian parameter $\nu(n)$.

**Example 1 (KT-SPLIT)** Given a kernel $\mathbf{k}$ and $n$ input points $\mathcal{S}_{\text{in}}$, the KT-SPLIT$(\delta)$ algorithm[1] of Dwivedi & Mackey (2022; 2021, Alg. 1a) takes $\Theta(n^2)$ kernel evaluations to output a coreset of size $n_{\text{out}}$ with better-than-i.i.d. integration error. Specifically, Dwivedi & Mackey (2022, Thm. 1) prove that, on an event with probability $1 - \frac{\delta}{2}$, KT-SPLIT$(\delta) \in \mathcal{G}^f(\nu)$ with

$$\nu(n) = \frac{2}{n_{\text{out}}\sqrt{3}}\sqrt{\log\left(\frac{6n_{\text{out}}\log_2(n/n_{\text{out}})}{\delta}\right)\|\mathbf{k}\|_\infty} \tag{2}$$

for all $f$ with $\|f\|_{\mathbf{k}} = 1$. ∎

Many algorithms also offer high-probability bounds on the kernel MMD (1), the worst-case integration error across the unit ball of the RKHS. We again capture these bounds abstractly using the following definition of a $\mathbf{k}$-sub-Gaussian thinning algorithm.

**Definition 3 (k-sub-Gaussian thinning algorithm)** *For a kernel $\mathbf{k}$, we call a thinning algorithm* ALG $\mathbf{k}$-sub-Gaussian *with parameter $v$ and shift $a$ and write* ALG $\in \mathcal{G}_{\mathbf{k}}(v, a)$ *if*

$$\mathbb{P}[\text{MMD}_{\mathbf{k}}(\mathcal{S}_{\text{in}}, \mathcal{S}_{\text{ALG}}) \geq a_n + v_n\sqrt{t} \,|\, \mathcal{S}_{\text{in}}] \leq e^{-t} \quad \text{for all} \quad t \geq 0. \tag{3}$$

*We also call $\varepsilon_{\mathbf{k},\text{ALG}}(n) \triangleq \max(v_n, a_n)$ the $\mathbf{k}$-sub-Gaussian error of* ALG.

**Example 2 (Kernel thinning)** Given a kernel $\mathbf{k}$ and $n$ input points $\mathcal{S}_{\text{in}}$, the generalized kernel thinning (KT$(\delta)$) algorithm[1] of Dwivedi & Mackey (2022; 2021, Alg. 1) takes $\Theta(n^2)$ kernel evaluations to output a coreset of size $n_{\text{out}}$ with near-optimal MMD error. In particular, by leveraging an appropriate auxiliary kernel $\mathbf{k}_{\text{split}}$, Dwivedi & Mackey (2022, Thms. 2-4) establish that, on an event with probability $1 - \frac{\delta}{2}$, KT$(\delta) \in \mathcal{G}_{\mathbf{k}}(a, v)$ with

$$a_n = \frac{C_a}{n_{\text{out}}}\sqrt{\|\mathbf{k}_{\text{split}}\|_\infty}, \quad \text{and} \quad v_n = \frac{C_v}{n_{\text{out}}}\sqrt{\|\mathbf{k}_{\text{split}}\|_\infty \log\left(\frac{6n_{\text{out}}\log_2(n/n_{\text{out}})}{\delta}\right)} \, \mathfrak{M}_{\mathcal{S}_{\text{in}},\mathbf{k}_{\text{split}}}, \tag{4}$$

where $\|\mathbf{k}_{\text{split}}\|_\infty = \sup_x \mathbf{k}_{\text{split}}(x, x)$, $C_a$ and $C_v$ are explicit constants, and $\mathfrak{M}_{\mathcal{S}_{\text{in}},\mathbf{k}_{\text{split}}} \geq 1$ is non-decreasing in $n$ and varies based on the tails of $\mathbf{k}_{\text{split}}$ and the radius of the ball containing $\mathcal{S}_{\text{in}}$. ∎

## 3 COMPRESS

The core subroutine of COMPRESS++ is a new meta-procedure called COMPRESS that, given a halving algorithm HALVE, an oversampling parameter $\mathfrak{g}$, and $n$ input points, outputs a thinned coreset of size $2^{\mathfrak{g}}\sqrt{n}$. The COMPRESS algorithm (Alg. 1) is very simple to implement: first, divide the input points into four subsequences of size $\frac{n}{4}$ (in any manner the user chooses); second, recursively call COMPRESS on each subsequence to produce four coresets of size $2^{\mathfrak{g}-1}\sqrt{n}$; finally, call HALVE on the concatenation of those coresets to produce the final output of size $2^{\mathfrak{g}}\sqrt{n}$. As we show in App. H, COMPRESS can also be implemented in a streaming fashion to consume at most $\mathcal{O}(4^{\mathfrak{g}}\sqrt{n})$ memory.

### 3.1 INTEGRATION ERROR AND RUNTIME GUARANTEES FOR COMPRESS

Our first result relates the runtime and single-function integration error of COMPRESS to the runtime and error of HALVE. We measure integration error for each function $f$ probabilistically in terms of the sub-Gaussian parameter $\nu$ of Def. 2 and measure runtime by the number of dominant operations performed by HALVE (e.g., the number of kernel evaluations performed by kernel thinning).

**Theorem 1 (Runtime and integration error of COMPRESS)** *If* HALVE *has runtime $r_{\text{H}}(n)$ for inputs of size $n$, then* COMPRESS *has runtime*

$$r_{\text{C}}(n) = \sum_{i=0}^{\beta_n} 4^i \cdot r_{\text{H}}(\ell_n 2^{-i}), \tag{5}$$

*where $\ell_n \triangleq 2^{\mathfrak{g}+1}\sqrt{n}$ (twice the output size of* COMPRESS*), and $\beta_n \triangleq \log_2(\frac{n}{\ell_n}) = \log_4 n - \mathfrak{g} - 1$. Furthermore, if, for some function $f$,* HALVE $\in \mathcal{G}^f(\nu_{\text{H}})$*, then* COMPRESS $\in \mathcal{G}^f(\nu_{\text{C}})$ *with*

$$\nu_{\text{C}}^2(n) = \sum_{i=0}^{\beta_n} 4^{-i} \cdot \nu_{\text{H}}^2(\ell_n 2^{-i}). \tag{6}$$

---

[1]The $\delta$ argument of KT-SPLIT$(\delta)$ or KT$(\delta)$ indicates that each parameter $\delta_i = \frac{\delta}{\ell}$ in Dwivedi & Mackey (2022, Alg. 1a), where $\ell$ is the size of the input point sequence compressed by KT-SPLIT$(\delta)$ or KT$(\delta)$.

---

**Algorithm 1:** COMPRESS

---

**Input:** halving algorithm HALVE, oversampling parameter $\mathfrak{g}$, point sequence $\mathcal{S}_{\text{in}}$ of size $n$

**if** $n = 4^{\mathfrak{g}}$ **then return** $\mathcal{S}_{\text{in}}$

Partition $\mathcal{S}_{\text{in}}$ into four arbitrary subsequences $\{\mathcal{S}_i\}_{i=1}^{4}$ each of size $n/4$

**for** $i = 1, 2, 3, 4$ **do**

   $\widetilde{\mathcal{S}}_i \leftarrow$ COMPRESS$(\mathcal{S}_i, \text{HALVE}, \mathfrak{g})$     // return coresets of size $2^{\mathfrak{g}} \cdot \sqrt{\frac{n}{4}}$

**end**

$\widetilde{\mathcal{S}} \leftarrow$ CONCATENATE$(\widetilde{\mathcal{S}}_1, \widetilde{\mathcal{S}}_2, \widetilde{\mathcal{S}}_3, \widetilde{\mathcal{S}}_4)$     // coreset of size $2 \cdot 2^{\mathfrak{g}} \cdot \sqrt{n}$

**return** HALVE$(\widetilde{\mathcal{S}})$     // coreset of size $2^{\mathfrak{g}}\sqrt{n}$

---

As we prove in App. B, the runtime guarantee (5) is immediate once we unroll the COMPRESS recursion and identify that COMPRESS makes $4^i$ calls to HALVE with input size $\ell_n 2^{-i}$. The error guarantee (6) is more subtle: here, COMPRESS benefits significantly from random cancellations among the *conditionally independent* and *mean-zero* HALVE errors. Without these properties, the errors from each HALVE call could compound without cancellation leading to a significant degradation in COMPRESS quality. Let us now unpack the most important implications of Thm. 1.

**Remark 1 (Near-linear runtime and quadratic speed-ups for COMPRESS)** Thm. 1 implies that a quadratic-time HALVE with $r_{\text{H}}(n) = n^2$ yields a near-linear time COMPRESS with $r_{\text{C}}(n) \leq 4^{\mathfrak{g}+1} n(\log_4(n) - \mathfrak{g})$. If HALVE instead has super-quadratic runtime $r_{\text{H}}(n) = n^{\tau}$, COMPRESS enjoys a quadratic speed-up: $r_{\text{C}}(n) \leq c'_{\tau} n^{\tau/2}$ for $c'_{\tau} \triangleq \frac{2^{\tau(\mathfrak{g}+2)}}{2^{\tau}-4}$. More generally, whenever HALVE has superlinear runtime $r_{\text{H}}(n) = n^{\tau} \rho(n)$ for some $\tau \geq 1$ and non-decreasing $\rho$, COMPRESS satisfies

$$r_{\text{C}}(n) \leq \begin{cases} c_{\tau} \cdot n \, (\log_4(n) - \mathfrak{g}) \, \rho(\ell_n) & \text{for } \tau \leq 2 \\ c'_{\tau} \cdot n^{\tau/2} \, \rho(\ell_n) & \text{for } \tau > 2 \end{cases} \quad \text{where} \quad c_{\tau} \triangleq 4^{(\tau-1)(\mathfrak{g}+1)}.$$

**Remark 2 (COMPRESS inflates sub-Gaussian error by at most $\sqrt{\log_4 n}$)** Thm. 1 also implies

$$\nu_{\text{C}}(n) \leq \sqrt{\beta_n + 1} \, \nu_{\text{H}}(\ell_n) = \sqrt{\log_4 n - \mathfrak{g}} \, \nu_{\text{H}}(\ell_n)$$

in the usual case that $n \, \nu_{\text{H}}(n)$ is non-decreasing in $n$. Hence the sub-Gaussian error of COMPRESS is at most $\sqrt{\log_4 n}$ larger than that of halving an input of size $\ell_n$. This is an especially strong benchmark, as $\ell_n$ is twice the output size of COMPRESS, and thinning from $n$ to $\frac{\ell_n}{2}$ points should incur at least as much approximation error as halving from $\ell_n$ to $\frac{\ell_n}{2}$ points.

**Example 3 (KT-SPLIT-COMPRESS)** Consider running COMPRESS with, for each HALVE input of size $\ell$, HALVE = KT-SPLIT$(\frac{\ell^2}{n4^{\mathfrak{g}+1}(\beta_n+1)}\delta)$ from Ex. 1. Since KT-SPLIT runs in time $\Theta(n^2)$, COMPRESS runs in near-linear $\mathcal{O}(n \log n)$ time by Rem. 1. In addition, as we detail in App. F.1, on an event of probability $1 - \frac{\delta}{2}$, every HALVE call invoked by COMPRESS is $f$-sub-Gaussian with

$$\nu_{\text{H}}(\ell) = \frac{4}{\ell\sqrt{3}} \sqrt{\log(\frac{12n4^{\mathfrak{g}}(\beta_n+1)}{\ell\delta})\|\mathbf{k}\|_{\infty}} \quad \text{for all } f \text{ with } \|f\|_{\mathbf{k}} = 1. \tag{7}$$

Hence, Rem. 2 implies that COMPRESS is $f$-sub-Gaussian on the same event with $\nu_{\text{C}}(n) \leq \sqrt{\log_4 n - \mathfrak{g}} \, \nu_{\text{H}}(\ell_n)$, a guarantee within $\sqrt{\log_4 n}$ of the original KT-SPLIT$(\delta)$ error (2). ∎

## 3.2 MMD GUARANTEES FOR COMPRESS

Next, we bound the MMD error of COMPRESS in terms of the MMD error of HALVE. Recall that $\text{MMD}_{\mathbf{k}}$ (1) represents the worst-case integration error across the unit ball of the RKHS of $\mathbf{k}$. Its proof, based on the concentration of subexponential matrix martingales, is provided in App. C.

**Theorem 2 (MMD guarantees for COMPRESS)** *Suppose* HALVE $\in \mathcal{G}_{\mathbf{k}}(a, v)$ *for* $n \, a_n$ *and* $n \, v_n$ *non-decreasing and* $\mathbb{E}\big[\mathbb{P}_{\text{HALVE}}\mathbf{k} \mid \mathcal{S}_{\text{in}}\big] = \mathbb{P}_{\text{in}}\mathbf{k}$. *Then* COMPRESS $\in \mathcal{G}_{\mathbf{k}}(\widetilde{a}, \widetilde{v})$ *with*

$$\widetilde{v}_n \triangleq 4(a_{\ell_n} + v_{\ell_n})\sqrt{2(\log_4 n - \mathfrak{g})}, \quad \text{and} \quad \widetilde{a}_n \triangleq \widetilde{v}_n \sqrt{\log(n+1)}, \tag{8}$$

*where* $\ell_n = 2^{\mathfrak{g}+1}\sqrt{n}$ *as in Thm. 1.*

**Remark 3 (Symmetrization)** We can convert any halving algorithm into one that satisfies the unbiasedness condition $\mathbb{E}[\mathbb{P}_{\text{HALVE}}\mathbf{k} \mid \mathcal{S}_{\text{in}}] = \mathbb{P}_{\text{in}}\mathbf{k}$ without impacting integration error by *symmetrization*, i.e., by returning either the outputted half or its complement with equal probability.

**Remark 4 (COMPRESS inflates MMD guarantee by at most $10\log(n+1)$)** Thm. 2 implies that the $\mathbf{k}$-sub-Gaussian error of COMPRESS is always at most $10\log(n+1)$ times that of HALVE with input size $\ell_n = 2^{\mathfrak{g}+1}\sqrt{n}$ since

$$\varepsilon_{\mathbf{k},\text{COMPRESS}}(n) \overset{\text{Def. 3}}{=} \max(\widetilde{a}_n, \widetilde{v}_n) \overset{(8)}{\leq} 10\log(n+1)\max(a_{\ell_n}, v_{\ell_n}) = 10\log(n+1)\cdot\varepsilon_{\mathbf{k},\text{HALVE}}(\ell_n).$$

As in Rem. 2, HALVE applied to an input of size $\ell_n$ is a particularly strong benchmark, as thinning from $n$ to $\frac{\ell_n}{2}$ points should incur at least as much MMD error as halving from $\ell_n$ to $\frac{\ell_n}{2}$.

**Example 4 (KT-COMPRESS)** Consider running COMPRESS with, for each HALVE input of size $\ell$, $\text{HALVE} = \text{KT}(\frac{\ell^2}{n4^{\mathfrak{g}+1}(\beta_n+1)}\delta)$ from Ex. 2 after symmetrizing as in Rem. 3. Since KT has $\Theta(n^2)$ runtime, COMPRESS yields near-linear $\mathcal{O}(n\log n)$ runtime by Rem. 1. Moreover, as we detail in App. F.2, using the notation of Ex. 2, on an event $\mathcal{E}$ of probability at least $1 - \frac{\delta}{2}$, every HALVE call invoked by COMPRESS is $\mathbf{k}$-sub-Gaussian with

$$a_\ell = \tfrac{2C_a}{\ell}\sqrt{\|\mathbf{k}\|_\infty}, \quad \text{and} \quad v_\ell = \tfrac{2C_v}{\ell}\sqrt{\|\mathbf{k}\|_\infty \log(\tfrac{12n4^{\mathfrak{g}}(\beta_n+1)}{\ell\delta})}\,\mathfrak{M}_{\mathcal{S}_{\text{in}},\mathbf{k}}.$$

Thus, Rem. 4 implies that, on $\mathcal{E}$, KT-COMPRESS has $\mathbf{k}$-sub-Gaussian error $\varepsilon_{\mathbf{k},\text{COMPRESS}}(n) \leq 10\log(n+1)\varepsilon_{\mathbf{k},\text{HALVE}}(\ell_n)$, a guarantee within $10\log(n+1)$ of the original KT$(\delta)$ MMD error (4). ∎

## 4 COMPRESS++

To offset any excess error due to COMPRESS while maintaining its near-linear runtime, we next introduce COMPRESS++ (Alg. 2), a simple two-stage meta-procedure for faster root-thinning. COMPRESS++ takes as input an oversampling parameter $\mathfrak{g}$, a halving algorithm HALVE, and a $2^{\mathfrak{g}}$-thinning algorithm THIN (see Def. 1). In our applications, HALVE and THIN are derived from the same base algorithm (e.g., from KT with different thinning factors), but this is not required. COMPRESS++ first runs the faster but slightly more erroneous COMPRESS(HALVE, $\mathfrak{g}$) algorithm to produce an intermediate coreset of size $2^{\mathfrak{g}}\sqrt{n}$. Next, the slower but more accurate THIN algorithm is run on the greatly compressed intermediate coreset to produce a final output of size $\sqrt{n}$. In the sequel, we demonstrate how to set $\mathfrak{g}$ to offset error inflation due to COMPRESS while maintaining its fast runtime.

---

**Algorithm 2: COMPRESS++**

**Input:** oversampling parameter $\mathfrak{g}$, halving alg. HALVE, $2^{\mathfrak{g}}$-thinning alg. THIN, point sequence $\mathcal{S}_{\text{in}}$ of size $n$

$\mathcal{S}_{\text{C}} \quad\leftarrow\quad \text{COMPRESS}(\text{HALVE}, \mathfrak{g}, \mathcal{S}_{\text{in}})$  // coreset of size $2^{\mathfrak{g}}\sqrt{n}$

$\mathcal{S}_{\text{C++}} \leftarrow\quad \text{THIN}(\mathcal{S}_{\text{C}})$  // coreset of size $\sqrt{n}$

**return** $\mathcal{S}_{\text{C++}}$

---

### 4.1 INTEGRATION ERROR AND RUNTIME GUARANTEES FOR COMPRESS++

The following result, proved in App. D, relates the runtime and single-function integration error of COMPRESS++ to the runtime and error of HALVE and THIN.

**Theorem 3 (Runtime and integration error of COMPRESS++)** *If* HALVE *and* THIN *have runtimes* $r_{\text{H}}(n)$ *and* $r_{\text{T}}(n)$ *respectively for inputs of size* $n$*, then* COMPRESS++ *has runtime*

$$r_{\text{C++}}(n) = r_{\text{C}}(n) + r_{\text{T}}(\ell_n/2) \quad \text{where} \quad r_{\text{C}}(n) \overset{(5)}{=} \sum_{i=0}^{\beta_n} 4^i \cdot r_{\text{H}}(\ell_n 2^{-i}), \tag{9}$$

$\ell_n = 2^{\mathfrak{g}+1}\sqrt{n}$, *and* $\beta_n = \log_4 n - \mathfrak{g} - 1$ *as in Thm. 1. Furthermore, if for some function* $f$, HALVE $\in \mathcal{G}^f(\nu_{\text{H}})$ *and* THIN $\in \mathcal{G}^f(\nu_{\text{T}})$*, then* COMPRESS++ $\in \mathcal{G}^f(\nu_{\text{C++}})$ *with*

$$\nu_{\text{C++}}^2(n) = \nu_{\text{C}}^2(n) + \nu_{\text{T}}^2(\ell_n/2) \quad \text{where} \quad \nu_{\text{C}}^2(n) \overset{(6)}{=} \sum_{i=0}^{\beta_n} 4^{-i} \cdot \nu_{\text{H}}^2(\ell_n 2^{-i}).$$

**Remark 5 (Near-linear runtime and near-quadratic speed-ups for COMPRESS++)** When HALVE and THIN have quadratic runtimes with $\max(r_{\mathrm{H}}(n), r_{\mathrm{T}}(n)) = n^2$, Thm. 3 and Rem. 1 yield that $r_{\mathrm{C++}}(n) \leq 4^{\mathfrak{g}+1}\, n(\log_4(n) - \mathfrak{g}) + 4^{\mathfrak{g}}n$. Hence, COMPRESS++ maintains a near-linear runtime

$$r_{\mathrm{C++}}(n) = \mathcal{O}(n\log_4^{c+1}(n)) \quad \text{whenever} \quad 4^{\mathfrak{g}} = \mathcal{O}(\log_4^c n). \tag{10}$$

If HALVE and THIN instead have super-quadratic runtimes with $\max(r_{\mathrm{H}}(n), r_{\mathrm{T}}(n)) = n^{\tau}$, then by Rem. 1 we have $r_{\mathrm{C++}}(n) \leq (\frac{4^{\tau}}{2^{\tau}-4} + 1)\, 2^{\mathfrak{g}\tau} n^{\tau/2}$, so that COMPRESS++ provides a near-quadratic speed up $r_{\mathrm{C++}}(n) = \mathcal{O}(n^{\tau/2}\log_4^{c\tau/2}(n))$ whenever $4^{\mathfrak{g}} = \mathcal{O}(\log_4^c n)$.

**Remark 6 (COMPRESS++ inflates sub-Gaussian error by at most $\sqrt{2}$)** In the usual case that $n\,\nu_{\mathrm{H}}(n)$ is non-decreasing in $n$, Thm. 3 and Rem. 2 imply that

$$\nu_{\mathrm{C++}}^2(n) \leq (\log_4 n - \mathfrak{g})\nu_{\mathrm{H}}^2(\ell_n) + \nu_{\mathrm{T}}^2(\tfrac{\ell_n}{2}) = \nu_{\mathrm{T}}^2(\tfrac{\ell_n}{2}) \cdot \left(1 + \tfrac{\log_4 n - \mathfrak{g}}{4^{\mathfrak{g}}} \cdot (\tfrac{\zeta_{\mathrm{H}}(\ell_n)}{\zeta_{\mathrm{T}}(\ell_n/2)})^2\right)$$

where we have introduced the rescaled quantities $\zeta_{\mathrm{H}}(\ell_n) \triangleq \frac{\ell_n}{2}\nu_{\mathrm{H}}(\ell_n)$ and $\zeta_{\mathrm{T}}(\tfrac{\ell_n}{2}) \triangleq \sqrt{n}\,\nu_{\mathrm{T}}(\tfrac{\ell_n}{2})$. Therefore, COMPRESS++ satisfies

$$\nu_{\mathrm{C++}}(n) \leq \sqrt{2}\nu_{\mathrm{T}}(\tfrac{\ell_n}{2}) \quad \text{whenever} \quad \mathfrak{g} \geq \log_4\log_4 n + \log_2(\tfrac{\zeta_{\mathrm{H}}(\ell_n)}{\zeta_{\mathrm{T}}(\ell_n/2)}). \tag{11}$$

That is, whenever COMPRESS++ is run with an oversampling parameter $\mathfrak{g}$ satisfying (11) its sub-Gaussian error is never more than $\sqrt{2}$ times the second-stage THIN error. Here, THIN represents a strong baseline for comparison as thinning from $\ell_n/2$ to $\sqrt{n}$ points should incur at least as much error as thinning from $n$ to $\sqrt{n}$ points.

As we illustrate in the next example, when THIN and HALVE are derived from the same thinning algorithm, the ratio $\frac{\zeta_{\mathrm{H}}(\ell_n)}{\zeta_{\mathrm{T}}(\ell_n/2)}$ is typically bounded by a constant $C$ so that the choice $\mathfrak{g} = \lceil\log_4\log_4 n + \log_2 C\rceil$ suffices to simultaneously obtain the $\sqrt{2}$ relative error guarantee (11) of Rem. 6 and the substantial speed-ups (10) of Rem. 5.

**Example 5 (KT-SPLIT-COMPRESS++)** In the notation of Ex. 1, consider running COMPRESS++ with HALVE = KT-SPLIT$(\frac{\ell^2}{4n2^{\mathfrak{g}}(\mathfrak{g}+2^{\mathfrak{g}}(\beta_n+1))}\delta)$ when applied to an input of size $\ell$ and THIN = KT-SPLIT$(\frac{\mathfrak{g}}{\mathfrak{g}+2^{\mathfrak{g}}(\beta_n+1)}\delta)$. As detailed in App. F.3, on an event of probability $1 - \frac{\delta}{2}$, all COMPRESS++ invocations of HALVE and THIN are simultaneously $f$-sub-Gaussian with parameters satisfying

$$\zeta_{\mathrm{H}}(\ell) = \zeta_{\mathrm{T}}(\ell) = \tfrac{2}{\sqrt{3}}\sqrt{\log(\tfrac{6\sqrt{n}(\mathfrak{g}+2^{\mathfrak{g}}(\beta_n+1))}{\delta})\|\mathbf{k}\|_{\infty}} \implies \tfrac{\zeta_{\mathrm{H}}(\ell_n)}{\zeta_{\mathrm{T}}(\frac{\ell_n}{2})} = 1 \text{ for all } f \text{ with } \|f\|_{\mathbf{k}} = 1. \tag{12}$$

Since KT-SPLIT runs in $\Theta(n^2)$ time, Rems. 5 and 6 imply that KT-SPLIT-COMPRESS++ with $\mathfrak{g} = \lceil\log_4\log_4 n\rceil$ runs in near-linear $\mathcal{O}(n\log^2 n)$ time and inflates sub-Gaussian error by at most $\sqrt{2}$. ∎

## 4.2 MMD GUARANTEES FOR COMPRESS++

Next, we bound the MMD error of COMPRESS++ in terms of the MMD error of HALVE and THIN. The proof of the following result can be found in App. E.

**Theorem 4 (MMD guarantees for COMPRESS++)** If THIN $\in \mathcal{G}_{\mathbf{k}}(a',v')$, HALVE $\in \mathcal{G}_{\mathbf{k}}(a,v)$ for $n\,a_n$ and $n\,v_n$ non-decreasing, and $\mathbb{E}[\mathbb{P}_{\mathrm{HALVE}}\mathbf{k} \mid \mathcal{S}_{\mathrm{in}}] = \mathbb{P}_{\mathrm{in}}\mathbf{k}$, then COMPRESS++ $\in \mathcal{G}_{\mathbf{k}}(\widehat{a}, \widehat{v})$ with

$$\widehat{v}_n \triangleq \widetilde{v}_n + v'_{\ell_n/2} \quad \text{and} \quad \widehat{a}_n \triangleq \widetilde{a}_n + a'_{\ell_n/2} + \widehat{v}_n\sqrt{\log 2}$$

for $\widetilde{v}_n$ and $\widetilde{a}_n$ defined in Thm. 2 and $\ell_n = 2^{\mathfrak{g}+1}\sqrt{n}$ as in Thm. 1.

**Remark 7 (COMPRESS++ inflates MMD guarantee by at most 4)** Thm. 4 implies that the COMPRESS++ $\mathbf{k}$-sub-Gaussian error $\varepsilon_{\mathbf{k},\mathrm{COMPRESS++}}(n) = \max(\widehat{a}_n, \widehat{v}_n)$ satisfies

$$\varepsilon_{\mathbf{k},\mathrm{COMPRESS++}}(n) \leq (10\log(n+1)\,\varepsilon_{\mathbf{k},\mathrm{HALVE}}(\ell_n) + \varepsilon_{\mathbf{k},\mathrm{THIN}}(\tfrac{\ell_n}{2}))\,(1 + \sqrt{\log 2})$$

$$\leq \varepsilon_{\mathbf{k},\mathrm{THIN}}(\tfrac{\ell_n}{2})(\tfrac{10\log(n+1)}{2^{\mathfrak{g}}}\tfrac{\widetilde{\zeta}_{\mathrm{H}}(\ell_n)}{\widetilde{\zeta}_{\mathrm{T}}(\frac{\ell_n}{2})} + 1)(1 + \sqrt{\log 2}),$$

where we have introduced the rescaled quantities $\widetilde{\zeta}_{\mathrm{H}}(\ell_n) \triangleq \frac{\ell_n}{2}\varepsilon_{\mathbf{k},\mathrm{HALVE}}(\ell_n)$ and $\widetilde{\zeta}_{\mathrm{T}}(\frac{\ell_n}{2}) \triangleq \sqrt{n}\,\varepsilon_{\mathbf{k},\mathrm{THIN}}(\frac{\ell_n}{2})$. Therefore, COMPRESS++ satisfies

$$\varepsilon_{\mathbf{k},\mathrm{COMPRESS++}}(n) \le 4\,\varepsilon_{\mathbf{k},\mathrm{THIN}}(\tfrac{\ell_n}{2}) \quad \text{whenever} \quad \mathfrak{g} \ge \log_2\log(n+1) + \log_2(8.5\tfrac{\widetilde{\zeta}_{\mathrm{H}}(\ell_n)}{\widetilde{\zeta}_{\mathrm{T}}(\frac{\ell_n}{2})}). \quad (13)$$

In other words, relative to a strong baseline of thinning from $\frac{\ell_n}{2}$ to $\sqrt{n}$ points, COMPRESS++ inflates $\mathbf{k}$-sub-Gaussian error by at most a factor of $4$ whenever $\mathfrak{g}$ satisfies (13). For example, when the ratio $\widetilde{\zeta}_{\mathrm{H}}(\ell_n)/\widetilde{\zeta}_{\mathrm{T}}(\frac{\ell_n}{2})$ is bounded by $C$, it suffices to choose $\mathfrak{g} = \lceil\log_2\log(n+1)+\log_2(8.5C)\rceil$.

**Example 6 (KT-COMPRESS++)** In the notation of Ex. 2 and Rem. 3, consider running COMPRESS++ with HALVE = symmetrized $\mathrm{KT}(\frac{\ell^2}{4n2^{\mathfrak{g}}(\mathfrak{g}+2^{\mathfrak{g}}(\beta_n+1))}\delta)$ when applied to an input of size $\ell$ and THIN = $\mathrm{KT}(\frac{\mathfrak{g}}{\mathfrak{g}+2^{\mathfrak{g}}(\beta_n+1)}\delta)$. As we detail in App. F.4, on an event of probability $1-\frac{\delta}{2}$, all COMPRESS++ invocations of HALVE and THIN are simultaneously $\mathbf{k}$-sub-Gaussian with

$$\widetilde{\zeta}_{\mathrm{H}}(\ell_n) = \widetilde{\zeta}_{\mathrm{T}}(\tfrac{\ell_n}{2}) = C_v\sqrt{\|\mathbf{k}\|_\infty \log(\tfrac{6\sqrt{n}(\mathfrak{g}+2^{\mathfrak{g}}(\beta_n+1))}{\delta})}\,\mathfrak{M}_{\mathcal{S}_{\mathrm{in}},\mathbf{k}} \quad \implies \quad \frac{\widetilde{\zeta}_{\mathrm{H}}(\ell_n)}{\widetilde{\zeta}_{\mathrm{T}}(\frac{\ell_n}{2})} = 1.$$

As KT runs in $\Theta(n^2)$ time, Rems. 5 and 7 imply that KT-COMPRESS++ with $\mathfrak{g} = \lceil\log_2\log n+3.1\rceil$ runs in near-linear $\mathcal{O}(n\log^3 n)$ time and inflates $\mathbf{k}$-sub-Gaussian error by at most 4. ∎

## 5 EXPERIMENTS

We now turn to an empirical evaluation of the speed-ups and error of COMPRESS++. We begin by describing the thinning algorithms, compression tasks, evaluation metrics, and kernels used in our experiments. Supplementary experimental details and results can be found in App. G.

**Thinning algorithms** Each experiment compares a high-accuracy, quadratic time thinning algorithm—either target kernel thinning (Dwivedi & Mackey, 2022) or kernel herding (Chen et al., 2010)—with our near-linear time COMPRESS and COMPRESS++ variants that use the same input algorithm to HALVE and THIN. In each case, we perform root thinning, compressing $n$ input points down to $\sqrt{n}$ points, so that COMPRESS is run with $\mathfrak{g} = 0$. For COMPRESS++, we use $\mathfrak{g} = 4$ throughout to satisfy the small relative error criterion (11) in all experiments. When halving we restrict each input algorithm to return distinct points and symmetrize the output as discussed in Rem. 3.

**Compressing i.i.d. summaries** To demonstrate the advantages of COMPRESS++ over equal-sized i.i.d. summaries we compress input point sequences $\mathcal{S}_{\mathrm{in}}$ drawn i.i.d. from either (a) Gaussian targets $\mathbb{P} = \mathcal{N}(0,\mathbf{I}_d)$ with $d \in \{2,4,10,100\}$ or (b) $M$-component mixture of Gaussian targets $\mathbb{P} = \frac{1}{M}\sum_{j=1}^M \mathcal{N}(\mu_j,\mathbf{I}_2)$ with $M \in \{4,6,8,32\}$ and component means $\mu_j \in \mathbb{R}^2$ defined in App. G.

**Compressing MCMC summaries** To demonstrate the advantages of COMPRESS++ over standard MCMC thinning, we also compress input point sequences $\mathcal{S}_{\mathrm{in}}$ generated by a variety of popular MCMC algorithms (denoted by RW, ADA-RW, MALA, and pMALA) targeting four challenging Bayesian posterior distributions $\mathbb{P}$. In particular, we adopt the four posterior targets of Riabiz et al. (2020a) based on the *Goodwin (1965) model* of oscillatory enzymatic control ($d = 4$), the *Lotka (1925); Volterra (1926) model* of oscillatory predator-prey evolution ($d = 4$), the *Hinch et al. (2004) model* of calcium signalling in cardiac cells ($d = 38$), and a tempered Hinch model posterior ($d = 38$). Notably, for the Hinch experiments, each summary point discarded via an accurate thinning procedure saves 1000s of downstream CPU hours by avoiding an additional critically expensive whole-heart simulation (Riabiz et al., 2020a). See App. G for MCMC algorithm and target details.

**Kernel settings** Throughout we use a Gaussian kernel $\mathbf{k}(x,y) = \exp(-\frac{1}{2\sigma^2}\|x-y\|_2^2)$ with $\sigma^2$ as specified by Dwivedi & Mackey (2021, Sec. K.2) for the MCMC targets and $\sigma^2 = 2d$ otherwise.

**Evaluation metrics** For each thinning procedure we report mean runtime across 3 runs and mean MMD error across 10 independent runs $\pm$ 1 standard error (the error bars are often too small to be visible). All runtimes were measured on a single core of an Intel Xeon CPU. For the i.i.d. targets, we report $\mathrm{MMD}_{\mathbf{k}}(\mathbb{P},\mathbb{P}_{\mathrm{out}})$ which can be exactly computed in closed-form. For the MCMC targets, we report the thinning error $\mathrm{MMD}_{\mathbf{k}}(\mathbb{P}_{\mathrm{in}},\mathbb{P}_{\mathrm{out}})$ analyzed directly by our theory (Thms. 2 and 4).

**Kernel thinning results** We first apply COMPRESS++ to the near-optimal KT algorithm to obtain comparable summaries at a fraction of the cost. Figs. 1 and 2 reveal that, in line with our guarantees,

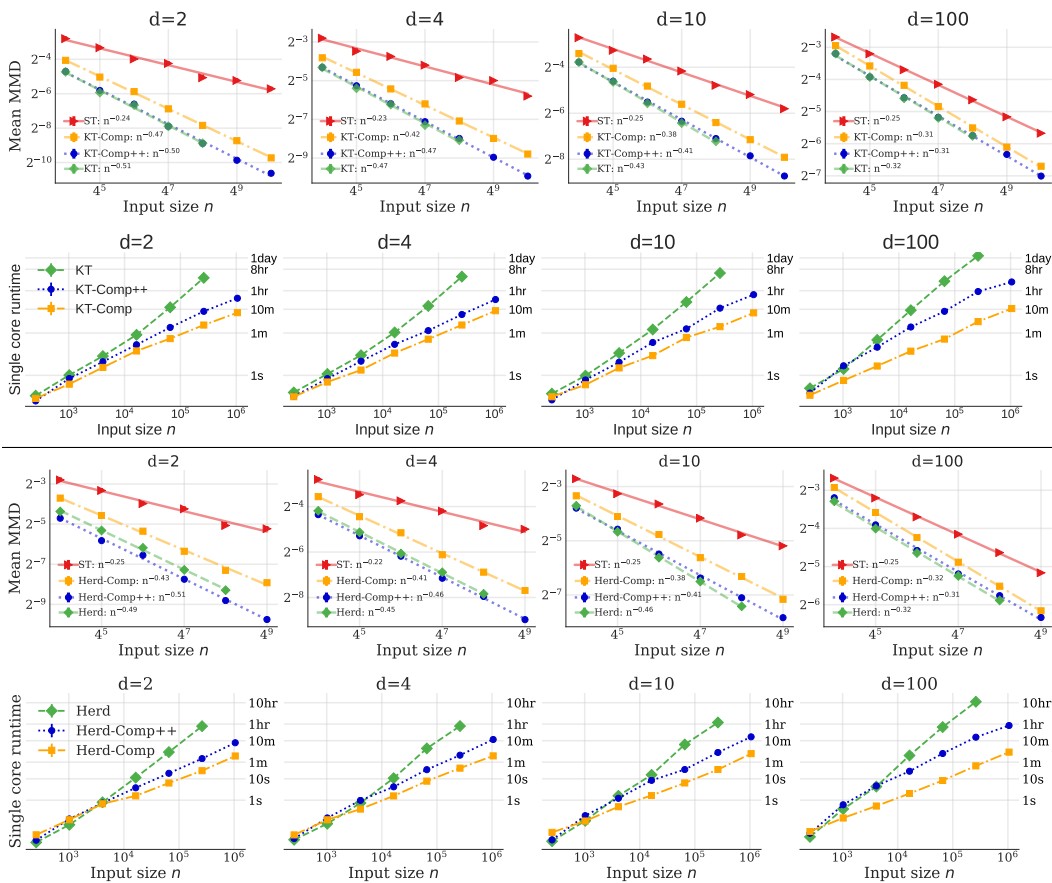

**Figure 1:** For Gaussian targets $\mathbb{P}$ in $\mathbb{R}^d$, KT-COMPRESS++ and Herd-COMPRESS++ improve upon the MMD of i.i.d. sampling (ST), closely track the error of their respective quadratic-time input algorithms KT and kernel herding (Herd), and substantially reduce the runtime.

KT-COMPRESS++ matches or nearly matches the MMD error of KT in all experiments while also substantially reducing runtime. For example, KT thins $65000$ points in $10$ dimensions in $20$m, while KT-COMPRESS++ needs only $1.5$m; KT takes more than a day to thin $250000$ points in $100$ dimensions, while KT-COMPRESS++ takes less than an hour (a $32\times$ speed-up). For reference we also display the error of standard thinning (ST) to highlight that KT-COMPRESS++ significantly improves approximation quality relative to the standard practice of i.i.d. summarization or standard MCMC thinning. See Fig. 4 in App. G.1 for analogous results with mixture of Gaussian targets.

**Kernel herding results** A strength of COMPRESS++ is that it can be applied to any thinning algorithm, including those with suboptimal or unknown performance guarantees that often perform well in practical. In such cases, Rems. 4 and 6 still ensure that COMPRESS++ error is never much larger than that of the input algorithm. As an illustration, we apply COMPRESS++ to the popular quadratic-time kernel herding algorithm (Herd). Fig. 1 shows that Herd-COMPRESS++ matches or nearly matches the MMD error of Herd in all experiments while also substantially reducing runtime. For example, Herd requires more than $11$ hours to compress $250000$ points in $100$ dimensions, while Herd-COMPRESS++ takes only $14$ minutes (a $45\times$ speed-up). Moreover, surprisingly, Herd-COMPRESS++ is consistently more accurate than the original kernel herding algorithm for lower dimensional problems. See Fig. 4 in App. G.1 for comparable results with mixture of Gaussian $\mathbb{P}$.

**Visualizing coresets** For a 32-component mixture of Gaussians target, Fig. 3 visualizes the coresets produced by i.i.d. sampling, KT, kernel herding, and their COMPRESS++ variants. The COMPRESS++ coresets closely resemble those of their input algorithms and, compared with i.i.d. sampling, yield visibly improved stratification across the mixture components.

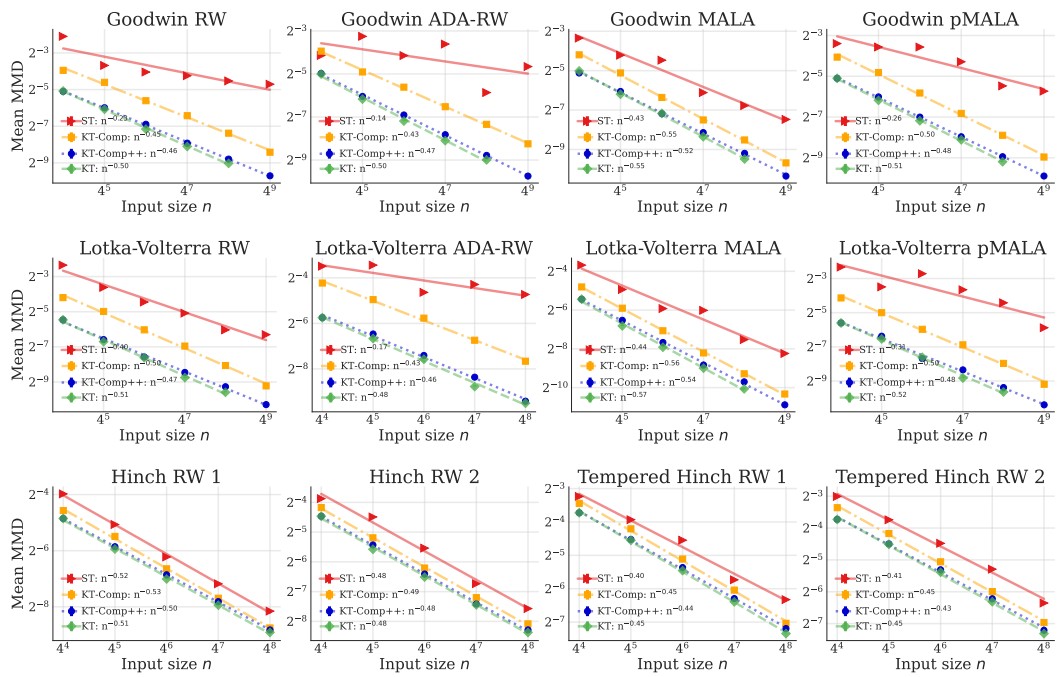

**Figure 2:** Given MCMC sequences summarizing challenging differential equation posteriors $\mathbb{P}$, KT-COMPRESS++ consistently improves upon the MMD of standard thinning (ST) and matches or nearly matches the error of of its quadratic-time input algorithm KT.

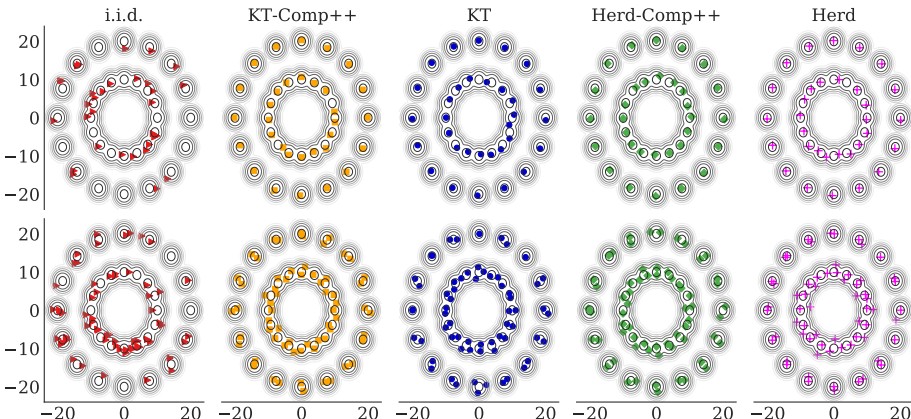

**Figure 3:** Coresets of size 32 (top) or 64 (bottom) with equidensity contours of the target underlaid.

## 6 DISCUSSION AND CONCLUSIONS

We introduced a new general meta-procedure, COMPRESS++, for speeding up thinning algorithms while preserving their error guarantees up to a factor of 4. When combined with the quadratic-time KT-SPLIT and kernel thinning algorithms of Dwivedi & Mackey (2021; 2022), the result is near-optimal distribution compression in near-linear time. Moreover, the same simple approach can be combined with any slow thinning algorithm to obtain comparable summaries in a fraction of the time. Two open questions recommend themselves for future investigation. First, why does Herd-COMPRESS++ improve upon the original kernel herding algorithm in lower dimensions, and can this improvement be extended to higher dimensions and to other algorithms? Second, is it possible to thin significantly faster than COMPRESS++ without significantly sacrificing approximation error? Lower bounds tracing out the computational-statistical trade-offs in distribution compression would provide a precise benchmark for optimality and point to any remaining opportunities for improvement.

REPRODUCIBILITY STATEMENT

See the `goodpoints` Python package for Python implementations of all methods in this paper and

https://github.com/microsoft/goodpoints

for code reproducing each experiment.

ACKNOWLEDGMENTS

We thank Carles Domingo-Enrich for alerting us that an outdated proof of Thm. 2 was previously included in the appendix. RD acknowledges the support by the National Science Foundation under Grant No. DMS-2023528 for the Foundations of Data Science Institute (FODSI). Part of this work was done when AS was interning at Microsoft Research New England.

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

## A ADDITIONAL DEFINITIONS AND NOTATION

This section provides additional definitions and notation used throughout the appendices.

We associate with each algorithm ALG and input $\mathcal{S}_{\mathrm{in}}$ the measure difference

$$\phi_{\mathrm{ALG}}(\mathcal{S}_{\mathrm{in}}) \triangleq \mathbb{P}_{\mathcal{S}_{\mathrm{in}}} - \mathbb{P}_{\mathcal{S}_{\mathrm{ALG}}} = \frac{1}{n}\sum_{x \in \mathcal{S}_{\mathrm{in}}} \delta_x - \frac{1}{n_{\mathrm{out}}}\sum_{x \in \mathcal{S}_{\mathrm{ALG}}} \delta_x \qquad (14)$$

that characterizes how well the output empirical distribution approximates the input. We will often write $\phi_{\mathrm{ALG}}$ instead of $\phi_{\mathrm{ALG}}(\mathcal{S}_{\mathrm{in}})$ for brevity if $\mathcal{S}_{\mathrm{in}}$ is clear from the context.

We also make use of the following standard definition of a sub-Gaussian random variable (see, e.g., Boucheron et al., 2013, Sec. 2.3).

**Definition 4 (Sub-Gaussian random variable)** *We say that a random variable $G$ is sub-Gaussian with parameter $\nu$ and write $G \in \mathcal{G}(\nu)$ if*

$$\mathbb{E}\big[\exp(\lambda\,G)\big] \leq \exp\Big(\tfrac{\lambda^2\nu^2}{2}\Big) \quad \textit{for all} \quad \lambda \in \mathbb{R}.$$

Given Def. 4, it follows that $\mathrm{ALG} \in \mathcal{G}^f(\nu)$ for a function $f$ as in Def. 2 if and only if the random variable $\phi_{\mathrm{ALG}}(f) \triangleq \mathbb{P}_{\mathcal{S}_{\mathrm{in}}} f - \mathbb{P}_{\mathcal{S}_{\mathrm{ALG}}} f$ is sub-Gaussian with parameter $\nu$ conditional on the input $\mathcal{S}_{\mathrm{in}}$.

In our proofs, it is often more convenient to work with an unnormalized *measure discrepancy*

$$\psi_{\mathrm{ALG}}(\mathcal{S}_{\mathrm{in}}) \triangleq n \cdot \phi_{\mathrm{ALG}}(\mathcal{S}_{\mathrm{in}}) \overset{(14)}{=} \sum_{x \in \mathcal{S}_{\mathrm{in}}} \delta_x - \frac{n}{n_{\mathrm{out}}}\sum_{\mathcal{S}_{\mathrm{ALG}}} \delta_x. \qquad (15)$$

By definition (15), we have the following useful equivalence:

$$\psi_{\mathrm{ALG}}(f) \triangleq n \cdot \phi_{\mathrm{ALG}}(f) \in \mathcal{G}(\sigma_{\mathrm{ALG}}) \iff \phi_{\mathrm{ALG}}(f) \in \mathcal{G}(\nu_{\mathrm{ALG}}) \quad \text{for} \quad \sigma_{\mathrm{ALG}} = n \cdot \nu_{\mathrm{ALG}}. \qquad (16)$$

The following standard lemma establishes that the sub-Gaussian property is closed under scaling and summation.

**Lemma 1 (Summation and scaling preserve sub-Gaussianity)** *Suppose $G_1 \in \mathcal{G}(\sigma_1)$. Then, for all $\beta \in \mathbb{R}$, we have $\beta \cdot G_1 \in \mathcal{G}(\beta\sigma_1)$. Furthermore, if $G_1$ is $\mathcal{F}$-measurable and $G_2 \in \mathcal{G}(\sigma_2)$ given $\mathcal{F}$, then $G_1 + G_2 \in \mathcal{G}(\sqrt{\sigma_1^2 + \sigma_2^2})$.*

**Proof** Fix any $\beta \in \mathbb{R}$. Since $G_1 \in \mathcal{G}(\sigma_1)$, for each $\lambda \in \mathbb{R}$,

$$\mathbb{E}\big[\exp(\lambda \cdot \beta \cdot G_1)\big] \leq \exp\Big(\tfrac{\lambda^2 (\beta \sigma_1)^2}{2}\Big),$$

so that $\beta G_1 \in \mathcal{G}(\beta \sigma_1)$ as advertised.

Furthermore, if $G_1$ is $\mathcal{F}$-measurable and $G_2 \in \mathcal{G}(\sigma_2)$ given $\mathcal{F}$, then, for each $\lambda \in \mathbb{R}$,

$$
\begin{aligned}
\mathbb{E}\Big[\exp\big(\lambda \cdot (G_1 + G_2)\big)\Big] = \mathbb{E}\big[\exp(\lambda \cdot G_1 + \lambda \cdot G_2)\big] &= \mathbb{E}\Big[\exp(\lambda \cdot G_1) \cdot \mathbb{E}\big[\exp(\lambda \cdot G_2) \mid \mathcal{F}\big]\Big] \\
&\leq \exp\Big(\tfrac{\lambda^2 \sigma_2^2}{2}\Big) \cdot \mathbb{E}\Big[\exp\big(\lambda \cdot f(G_2)\big)\Big] \\
&= \exp\Big(\tfrac{\lambda^2 \sigma_1^2}{2}\Big) \cdot \exp\Big(\tfrac{\lambda^2 \sigma_2^2}{2}\Big) = \exp\Big(\tfrac{\lambda^2 (\sigma_1^2 + \sigma_2^2)}{2}\Big),
\end{aligned}
$$

so that $G_1 + G_2 \in \mathcal{G}(\sqrt{\sigma_1^2 + \sigma_2^2})$ as claimed. $\qquad\square$

## B  PROOF OF THM. 1: RUNTIME AND INTEGRATION ERROR OF COMPRESS

First, we bound the running time of COMPRESS. By definition, COMPRESS makes four recursive calls to COMPRESS on inputs of size $n/4$. Then, HALVE is run on an input of size $2^{\mathfrak{g}+1}\sqrt{n}$. Thus, $r_{\mathrm{C}}$ satisfies the recursion

$$r_{\mathrm{C}}(n) = 4r_{\mathrm{C}}\big(\tfrac{n}{4}\big) + r_{\mathrm{H}}(\sqrt{n}2^{\mathfrak{g}+1}).$$

Since $r_{\mathrm{C}}(4^{\mathfrak{g}}) = 0$, we may unroll the recursion to find that

$$r_{\mathrm{C}}(n) = \textstyle\sum_{i=0}^{\beta_n} 4^i r_{\mathrm{H}}(2^{\mathfrak{g}+1}\sqrt{n4^{-i}}),$$

as claimed in (5).

Next, we bound the sub-Gaussian error for a fixed function $f$. In the measure discrepancy (15) notation of App. A we have

$$\psi_{\mathrm{C}}(\mathcal{S}_{\mathrm{in}}) = \textstyle\sum_{i=1}^4 \psi_{\mathrm{C}}(\mathcal{S}_i) + \sqrt{n}2^{-\mathfrak{g}-1}\psi_{\mathrm{H}}(\widetilde{\mathcal{S}}) \tag{17}$$

where $\mathcal{S}_i$ and $\widetilde{\mathcal{S}}$ are defined as in Alg. 1. Unrolling this recursion, we find that running COMPRESS on an input of size $n$ with oversampling parameter $\mathfrak{g}$ leads to applying HALVE on $4^i$ coresets of size $n_i = 2^{\mathfrak{g}+1-i}\sqrt{n}$ for $0 \leq i \leq \beta_n$. Denoting these HALVE inputs by $(\mathcal{S}_{i,j}^{\mathrm{in}})_{j \in [4^i]}$, we have

$$\psi_{\mathrm{C}}(\mathcal{S}_{\mathrm{in}}) = \sqrt{n}2^{-\mathfrak{g}-1} \textstyle\sum_{i=0}^{\beta_n} \sum_{j=1}^{4^i} 2^{-i}\psi_{\mathrm{H}}(\mathcal{S}_{i,j}^{\mathrm{in}}). \tag{18}$$

Now define $\sigma_{\mathrm{H}}(n) = n\nu_{\mathrm{H}}(n)$. Since $\psi_{\mathrm{H}}(\mathcal{S}_{i,j}^{\mathrm{in}})(f)$ are $\sigma_{\mathrm{H}}(n_i)$ sub-Gaussian given $(\mathcal{S}_{i',j'}^{\mathrm{in}})_{i'>i,j'\geq 1}$ and $(\mathcal{S}_{i,j'}^{\mathrm{in}})_{j'\leq j}$, Lem. 1 implies that $\psi_{\mathrm{C}}(\mathcal{S}_{\mathrm{in}})(f)$ is $\sigma_{\mathrm{C}}$ sub-Gaussian given $\mathcal{S}_{\mathrm{in}}$ for

$$\sigma_{\mathrm{C}}^2(n) = n4^{-\mathfrak{g}-1} \textstyle\sum_{i=0}^{\beta_n} \sigma_{\mathrm{H}}^2(n_i).$$

Recalling the relation (16) between $\sigma$ and $\nu$ from App. A, we conclude that

$$\nu_{\mathrm{C}}^2(n) = \textstyle\sum_{i=0}^{\beta_n} 4^{-i}\nu_{\mathrm{H}}^2(n_i).$$

as claimed in (6).

## C  PROOF OF THM. 2: MMD GUARANTEES FOR COMPRESS

Our proof proceeds in several steps. To control the MMD (1), we will control the Hilbert norm of the measure discrepancy of COMPRESS (15), which we first write as a weighted sum of measure discrepancies from different (conditionally independent) runs of HALVE. To effectively leverage the MMD tail bound assumption for this weighted sum, we reduce the problem to establishing a

concentration inequality for the operator norm of an associated matrix. We carry out this plan in four steps summarized below.

First, in App. C.1 we express the MMD associated with each HALVE measure discrepancy as the Euclidean norm of a suitable vector (Lem. 2). Second, in App. C.2 we define a matrix dilation operator for a vector that allows us to control vector norms using matrix spectral norms (Lem. 3). Third, in App. C.3 we prove and apply a sub-Gaussian matrix Freedman concentration inequality (Lem. 4) to control the MMD error for the COMPRESS output, which in turn requires us to establish moment bounds for these matrices by leveraging tail bounds for the MMD error (Lem. 5). Finally, we put together the pieces in App. C.4 to complete the proof.

We now begin our formal argument. We will make use of the unrolled representation (17) for the COMPRESS measure discrepancy $\psi_\text{C}(\mathcal{S}_\text{in})$ in terms of the HALVE inputs $(\mathcal{S}_{k,j}^\text{in})_{j \in [4^k]}$ of size $n_k = 2^{\mathfrak{g}+1-k}\sqrt{n}$ for $0 \le k \le \log_4 n - \mathfrak{g} - 1$. For brevity, we will use the shorthand $\psi_\text{C} \triangleq \psi_\text{C}(\mathcal{S}_\text{in})$, $\psi_{k,j}^\text{H} \triangleq \psi_\text{H}(\mathcal{S}_{k,j}^\text{in})$, and $\psi_\text{T} \triangleq \psi_\text{T}(\mathcal{S}_\text{C})$ hereafter.

## C.1 REDUCING MMD TO VECTOR EUCLIDEAN NORM

Number the elements of $\mathcal{S}_\text{in}$ as $(x_1, \ldots, x_n)$, define the $n \times n$ kernel matrix $\mathbf{K} \triangleq (\mathbf{k}(x_i, x_j))_{i,j=1}^n$, and let $\mathbf{K}^{\frac{1}{2}}$ denote a matrix square-root such that $\mathbf{K} = \mathbf{K}^{\frac{1}{2}} \cdot \mathbf{K}^{\frac{1}{2}}$ (which exists since $\mathbf{K}$ is a positive semidefinite matrix for any kernel $\mathbf{k}$). Next, let $\mathcal{S}_{k,j}^\text{out}$ denote the output sequence corresponding to $\psi_{k,j}^\text{H}$ (i.e., running HALVE on $\mathcal{S}_{k,j}^\text{in}$), and let $\{e_i\}_{i=1}^n$ denote the canonical basis of $\mathbb{R}^n$. The next lemma (with proof in App. C.5) relates the Hilbert norms to Euclidean norms of carefully constructed vectors.

**Lemma 2 (MMD as a vector norm)** *Define the vectors*

$$u_{k,j} \triangleq \mathbf{K}^{\frac{1}{2}} \sum_{i=1}^n e_i \Big( \mathbf{1}(x_i \in \mathcal{S}_{k,j}^\text{in}) - 2 \cdot \mathbf{1}(x_i \in \mathcal{S}_{k,j}^\text{out}) \Big), \text{ and } u_\text{C} \triangleq \sum_{k=0}^{\log_4 n - \mathfrak{g} - 1} \sum_{j=1}^{4^k} w_{k,j} u_{k,j}, \quad (19)$$

*where $w_{k,j} \triangleq \frac{\sqrt{n}}{2^{\mathfrak{g}+1+k}}$. Then, we have*

$$n^2 \cdot \text{MMD}_\mathbf{k}^2(\mathcal{S}_\text{in}, \mathcal{S}_\text{C}) = \|u_\text{C}\|_2^2, \quad \text{and} \quad (20)$$

$$\mathbb{E}[u_{k,j} | (u_{k',j'} : j' \in [4^{k'}], k' > k)] = 0 \quad \text{for} \quad k = 0, \ldots, \log_4 n - \mathfrak{g} - 2, \quad (21)$$

*and $u_{k,j}$ for $j \in [4^k]$ are conditionally independent given $(u_{k',j'} : j' \in [4^{k'}], k' > k)$.*

Applying (20), we effectively reduce the task of controlling the MMD errors to controlling the Euclidean norm of suitably defined vectors. Next, we reduce the problem to controlling the spectral norm of a suitable matrix.

## C.2 REDUCING VECTOR EUCLIDEAN NORM TO MATRIX SPECTRAL NORM

To this end, we define a symmetric dilation matrix operator: given a vector $u \in \mathbb{R}^n$, define the matrix $\mathbf{M}_u$ as

$$\mathbf{M}_u \triangleq \begin{pmatrix} 0 & u^\top \\ u & \mathbf{0}_{n \times n} \end{pmatrix} \in \mathbb{R}^{(n+1) \times (n+1)}. \quad (22)$$

It is straightforward to see that $u \mapsto \mathbf{M}_u$ is a linear map. In addition, the matrix $\mathbf{M}_u$ also satisfies a few important properties (established in App. C.6) that we use in our proofs.

**Lemma 3 (Properties of the dilation operator)** *For any $u \in \mathbb{R}^n$, the matrix $\mathbf{M}_u$ (22) satisfies*

$$\|\mathbf{M}_u\|_\text{op} \overset{(a)}{=} \|u\|_2 \overset{(b)}{=} \lambda_\text{max}(\mathbf{M}_u), \quad \text{and} \quad \mathbf{M}_u^q \overset{(c)}{\preceq} \|u\|_2^q \mathbf{I}_{n+1} \text{ for all } q \in \mathbb{N}. \quad (23)$$

Define the shorthand $\mathbf{M}_{k,j} \triangleq \mathbf{M}_{w_{k,j} u_{k,j}}$ (defined in Lem. 2). Applying Lems. 2 and 3, we find that

$$n \, \text{MMD}_\mathbf{k}(\mathcal{S}_\text{in}, \mathcal{S}_\text{C}) \overset{(20)}{=} \|u_\text{C}\|_2 \overset{(23)}{=} \lambda_\text{max}(\mathbf{M}_{u_\text{C}}) \overset{(i)}{=} \lambda_\text{max}\Big(\sum_{k=0}^{\log_4 n - \mathfrak{g} - 1} \sum_{j=1}^{4^k} \mathbf{M}_{k,j}\Big), \quad (24)$$

where step (i) follows from the linearity of the dilation operator. Thus to control the MMD error, it suffices to control the maximum eigenvalue of the sum of matrices appearing in (24).

## C.3 CONTROLLING THE SPECTRAL NORM VIA A SUB-GAUSSIAN MATRIX FREEDMAN INEQUALITY

To control the maximum eigenvalue of the matrix $\mathbf{M}_{u^c}$, we make use of (24) and the following sub-Gaussian generalization of the matrix Freedman inequality of Tropp (2012, Thm. 7.1). The proof of Lem. 4 can be found in App. C.7. For two matrices $A$ and $B$ of the same size, we write $A \preceq B$ if $B - A$ is positive semidefinite.

**Lemma 4 (Sub-Gaussian matrix Freedman inequality)** *Consider a sequence $(\mathbf{Y}_i)_{i=1}^N$ of self-adjoint random matrices in $\mathbb{R}^{m \times m}$ and a fixed sequence of scalars $(R_i)_{i=1}^N$ satisfying*

$$\mathbb{E}\left[\mathbf{Y}_i | (\mathbf{Y}_j)_{j=1}^{i-1}\right] \overset{(A)}{=} 0 \;\; and \;\; \mathbb{E}\left[\mathbf{Y}_i^q | (\mathbf{Y}_j)_{j=1}^{i-1}\right] \overset{(B)}{\preceq} (\tfrac{q}{2})! R_i^q \mathbf{I}, \; \text{for all } i \in [N] \text{ and } q \in 2\mathbb{N}. \quad (25)$$

*Define the variance parameter $\sigma^2 \triangleq \sum_{i=1}^N R_i^2$. Then,*

$$\mathbb{P}[\lambda_{\max}(\textstyle\sum_{i=1}^N \mathbf{Y}_i) \geq \sigma\sqrt{8(t + \log m)}] \leq e^{-t} \quad \text{for all} \quad t > 0,$$

*and equivalently*

$$\mathbb{P}[\lambda_{\max}(\textstyle\sum_{i=1}^N \mathbf{Y}_i) \leq \sigma\sqrt{8\log(m/\delta)}] \geq 1 - \delta \quad \text{for all} \quad \delta \in (0, 1].$$

To apply Lem. 4 with the matrices $\mathbf{M}_{k,j}$, we need to establish the zero-mean and moment bound conditions for suitable $R_{k,j}$ in (25).

### C.3.1 VERIFYING THE ZERO MEAN CONDITION (25)(A) FOR $\mathbf{M}_{k,j}$

To this end, first we note that the conditional independence and zero-mean property of $\psi_{k,j}^{\mathrm{H}}$ implies that the random vectors $u_{k,j}$ and the matrices $\mathbf{M}_{k,j}$ also satisfy a similar property, and in particular that

$$\mathbb{E}\left[\mathbf{M}_{k,j} \mid \left(\mathbf{M}_{k',j'} : k' > k, j' \in [4^{k'}]\right)\right] = \mathbf{0} \quad \text{for} \quad j \in [4^k], k \in \{0, 1, \ldots, \log_4 n - \mathfrak{g} - 1\}. \quad (26)$$

### C.3.2 ESTABLISHING MOMENT BOUND CONDITIONS (25)(B) FOR $\mathbf{M}_{k,j}$ IN TERMS OF $R_{k,j}$ VIA MMD TAIL BOUNDS FOR HALVE

To establish the moment bounds on $\mathbf{M}_{k,j}$, note that Lems. 2 and 3 imply that

$$\mathbf{M}_{k,j}^q = \mathbf{M}_{w_{k,j} u_{k,j}}^q \overset{(23)}{\preceq} \left\| w_{k,j} u_{k,j} \right\|_2^q \cdot \mathbf{I}_{n+1} \overset{(20)}{=} w_{k,j}^q \left\| u_{k,j} \right\|_2^q \cdot \mathbf{I}_{n+1} \quad (27)$$

where $w_{k,j}$ was defined in Lem. 2. Thus it suffices to establish the moment bounds on $\left\| u_{k,j} \right\|_2^q$. To this end, we first state a lemma that converts tail bounds to moment bounds. See App. C.8 for the proof inspired by Boucheron et al. (2013, Thm. 2.3).

**Lemma 5 (Tail bounds imply moment bounds)** *For a non-negative random variable $Z$,*

$$\mathbb{P}[Z > a + v\sqrt{t}] \leq e^{-t} \text{ for all } t \geq 0 \implies \mathbb{E}[Z^q] \leq (2a + 2v)^q (\tfrac{q}{2})! \text{ for all } q \in 2\mathbb{N}.$$

To obtain a moment bound for $\left\| u_{k,j} \right\|_2$, we first state some notation. For each $n$, define the quantities

$$a'_n \triangleq n a_n, \quad v'_n \triangleq n v_n \quad (28)$$

where $a_n$ and $v_n$ are the parameters such that HALVE $\in \mathcal{G}_{\mathbf{k}}(a_n, v_n)$ on inputs of size $n$. Using an argument similar to Lem. 2, we have

$$\left\| u_{k,j} \right\|_2 = n_{k,j} \operatorname{MMD}_{\mathbf{k}}(\mathcal{S}_{k,j}^{\mathrm{in}}, \mathcal{S}_{k,j}^{\mathrm{out}}) \quad \text{for} \quad n_{k,j} = |\mathcal{S}_{k,j}^{\mathrm{in}}| = \sqrt{n} 2^{\mathfrak{g}+1-k}.$$

Thereby, using the $\mathcal{G}_{\mathbf{k}}$ assumption on HALVE implies that

$$\mathbb{P}[\left\| u_{k,j} \right\|_2 \geq a'_{\ell'_k} + v'_{\ell'_k} \sqrt{t} \mid (u_{k',j'} : j' \in [4^{k'}], k' > k)] \leq e^{-t} \text{ for all } t \geq 0, \quad (29)$$

where

$$\ell_k' \triangleq n_{k,j} = \sqrt{n}2^{\mathfrak{g}+1-k} \tag{30}$$

and, notably, $\ell_n = \ell_0'$. Combining the bound (29) with Lem. 5 yields that

$$\mathbb{E}[\|u_{k,j}\|_2^q \mid (u_{k',j'} : j' \in [4^{k'}], k' > k)] \leq (\tfrac{q}{2})!(2a_{\ell_k'}' + 2v_{\ell_k'}')^q, \tag{31}$$

for all $q \in 2\mathbb{N}$, where $\ell_k$ is defined in (29). Now, putting together (27) and (31), and using the conditional independence of $\mathbf{M}_{k,j}$, we obtain the following control on the $q$-th moments of $\mathbf{M}_{k,j}$ for $q \in 2\mathbb{N}$:

$$\mathbb{E}\left[\mathbf{M}_{k,j}^q \mid \left(\mathbf{M}_{k',j'}, k' > k, j' \in [4^{k'}]\right)\right] \overset{(27)}{\preceq} w_{k,j}^q \cdot \mathbb{E}\left[\|u_{k,j}\|_2^q \mid \left\{u_{k',j'}, k' > k, j' \in [4^{k'}]\right\}\right] \cdot \mathbf{I}_{n+1}$$

$$\overset{(31)}{\preceq} w_{k,j}^q \cdot \left((2a_{\ell_k'}' + 2v_{\ell_k'}')^q (\tfrac{q}{2}!)\right) \cdot \mathbf{I}_{n+1}$$

$$= (\tfrac{q}{2})! R_{k,j}^q \mathbf{I}_{n+1} \text{ where } R_{k,j} \triangleq 2w_{k,j}(a_{\ell_k'}' + v_{\ell_k'}') \tag{32}$$

where $\ell_k$ is defined in (30). In summary, the computation above establishes the condition (B) from the display (25) for the matrices $\mathbf{M}_{k,j}$ in terms of the sequence $R_{k,j}$ defined in (32).

## C.4 Putting the pieces together for proving Thm. 2

Define

$$\widetilde{\sigma} \triangleq \sqrt{\log_4 n - \mathfrak{g}} \cdot 2(a_{\sqrt{n}2^{\mathfrak{g}+1}} + v_{\sqrt{n}2^{\mathfrak{g}+1}}) \tag{33}$$

Now, putting (26) and (32) together, we conclude that with a suitable ordering of the indices $(k,j)$, the assumptions of Lem. 4 are satisfied by the random matrices $\left(\mathbf{M}_{k,j}, j \in [4^k], k \in \{0, 1, \dots, \log_4 n - \mathfrak{g} - 1\}\right)$ with the sequence $\left(R_{k,j}\right)$. Now, since $\ell_k' = \sqrt{n}2^{\mathfrak{g}+1-k}$ (29) is decreasing in $k$, $w_{k,j} = \frac{\ell_k'}{4^{\mathfrak{g}+1}}$ (as defined in Lem. 2), and $a_n'$ and $v_n'$ (28) are assumed non-decreasing in $n$, we find that

$$n^2 \cdot \widetilde{\sigma}^2 \overset{(33)}{=} n^2(\log_4 n - \mathfrak{g})(2(a_{\sqrt{n}2^{\mathfrak{g}+1}} + v_{\sqrt{n}2^{\mathfrak{g}+1}}))^2$$

$$\overset{(29)}{=} (\log_4 n - \mathfrak{g})\frac{n}{4^{\mathfrak{g}+1}}(2(a_{\ell_0'}' + v_{\ell_0'}'))^2$$

$$\geq \sum_{k=0}^{\log_4 n - \mathfrak{g} - 1} \frac{n}{4^{\mathfrak{g}+1}}(2(a_{\ell_k'}' + v_{\ell_k'}'))^2$$

$$= \sum_{k=0}^{\log_4 n - \mathfrak{g} - 1} \sum_{j=1}^{4^k} \frac{n}{4^{\mathfrak{g}+1+k}}(2(a_{\ell_k'}' + v_{\ell_k'}'))^2$$

$$= \sum_{k=0}^{\log_4 n - \mathfrak{g} - 1} \sum_{j=1}^{4^k} (2w_{k,j}(a_{\ell_k'}' + v_{\ell_k'}'))^2$$

$$\overset{(32)}{=} \sum_{k=0}^{\log_4 n - \mathfrak{g} - 1} \sum_{j=1}^{4^k} R_{k,j}^2.$$

Finally, applying (24) and invoking Lem. 4 with $\sigma \leftarrow n\widetilde{\sigma}$ and $m \leftarrow n + 1$, we conclude that

$$\mathbb{P}[\mathrm{MMD}(\mathcal{S}_{\mathrm{in}}, \mathcal{S}_{\mathrm{C}}) \geq \widetilde{\sigma}\sqrt{8(\log(n+1) + t)}]$$

$$\overset{(24)}{=} \mathbb{P}[\lambda_{\max}(\sum_{k=0}^{\log_4 n - \mathfrak{g} - 1} \sum_{j=1}^{4^k} \mathbf{M}_{k,j}) \geq n\widetilde{\sigma}\sqrt{8(\log(n+1) + t)}]$$

$$\leq e^{-t} \quad \text{for all} \quad t > 0,$$

which in turn implies

$$\mathbb{P}[\mathrm{MMD}(\mathcal{S}_{\mathrm{in}}, \mathcal{S}_{\mathrm{C}}) \geq \widetilde{a}_n + \widetilde{v}_n\sqrt{t}] \leq e^{-t} \text{ for } t \geq 0,$$

since the parameters $\widetilde{v}_n, \widetilde{a}_n$ (8) satisfy

$$\widetilde{v}_n \overset{(8)}{=} 4(a_{\ell_n} + v_{\ell_n})\sqrt{2(\log_4 n - \mathfrak{g})} \overset{(33)}{=} \widetilde{\sigma}\sqrt{8}, \quad \text{and} \quad \widetilde{a}_n \overset{(8)}{=} \widetilde{v}_n\sqrt{\log(n+1)} = \widetilde{\sigma}\sqrt{8\log(n+1)}.$$

Comparing with Def. 3, Thm. 2 follows.

## C.5 PROOF OF LEM. 2: MMD AS A VECTOR NORM

Let $v_{k,j} \triangleq \sum_{i=1}^{n} e_i \Big( \mathbf{1}(x_i \in \mathcal{S}_{k,j}^{\text{in}}) - 2 \cdot \mathbf{1}(x_i \in \mathcal{S}_{k,j}^{\text{out}}) \Big)$. By the reproducing property of $\mathbf{k}$ we have

$$
\begin{aligned}
\|\psi_{k,j}^{\text{H}}(\mathbf{k})\|_{\mathbf{k}}^2 &= \Big\| \sum_{x \in \mathcal{S}_{k,j}^{\text{in}}} \mathbf{k}(x, \cdot) - 2 \sum_{x \in \mathcal{S}_{k,j}^{\text{out}}} \mathbf{k}(x, \cdot) \Big\|_{\mathbf{k}}^2 \\
&= \sum_{x \in \mathcal{S}_{k,j}^{\text{in}}, y \in \mathcal{S}_{k,j}^{\text{in}}} \mathbf{k}(x, y) - 2 \sum_{x \in \mathcal{S}_{k,j}^{\text{out}}, y \in \mathcal{S}_{k,j}^{\text{in}}} \mathbf{k}(x, y) + \sum_{x \in \mathcal{S}_{k,j}^{\text{out}}, y \in \mathcal{S}_{k,j}^{\text{out}}} \mathbf{k}(x, y) \\
&= v_{k,j}^\top \mathbf{K} v_{k,j} \stackrel{(19)}{=} \|u_{k,j}\|_2^2.
\end{aligned}
\tag{34}
$$

Using (18), (19), and (22), and mimicking the derivation above (34), we can also conclude that

$$
\|\psi_{\text{C}}(\mathbf{k})\|_{\mathbf{k}}^2 = \|u_{\text{C}}\|_2^2.
$$

Additionally, we note that

$$
\text{MMD}_{\mathbf{k}}(\mathcal{S}_{\text{in}}, \mathcal{S}_{\text{C}}) = \sup_{\|f\|_{\mathbf{k}}=1} \tfrac{1}{n} \langle f, \psi_{\text{C}}(\mathbf{k}) \rangle_{\mathcal{H}_{\mathbf{k}}} = \tfrac{1}{n} \|\psi_{\text{C}}(\mathbf{k})\|_{\mathbf{k}}.
$$

Finally the conditional independence and zero mean property (21) follows from (18) by noting that conditioned on $(\mathcal{S}_{k',j'}^{\text{in}})_{k'>k,j'\geq 1}$, the sets $(\mathcal{S}_{k,j}^{\text{in}})_{j\geq 1}$ are independent.

## C.6 PROOF OF LEM. 3: PROPERTIES OF THE DILATION OPERATOR

For claim (a) in the display (23), we have

$$
\mathbf{M}_u^2 = \begin{pmatrix} \|u\|_2^2 & \mathbf{0}_n^\top \\ \mathbf{0}_n & uu^\top \end{pmatrix} \stackrel{(i)}{\preceq} \|u\|_2^2 \mathbf{I}_{n+1} \implies \|\mathbf{M}_u\|_{\text{op}} \stackrel{(ii)}{=} \|u\|_2,
$$

where step (i) follows from the standard fact that $uu^\top \preceq \|u\|_2^2 \mathbf{I}_n$ and step (ii) from the facts $\mathbf{M}_u^2 \widetilde{e}_1 = \|u\|_2^2 \widetilde{e}_1$ for $\widetilde{e}_1$ the first canonical basis vector of $\mathbb{R}^{n+1}$ and $\|\mathbf{M}_u\|_{\text{op}}^2 = \|\mathbf{M}_u^2\|_{\text{op}}$. Claim (b) follows directly by verifying that the vector $v = [1, \frac{u^\top}{\|u\|_2}]^\top$ is an eigenvector of $\mathbf{M}_u$ with eigenvalue $\|u\|_2$. Finally, claim (c) follows directly from the claim (a) and the fact that $\|\mathbf{M}_u^q\|_{\text{op}} = \|\mathbf{M}_u\|_{\text{op}}^q$ for all integers $q \geq 1$.

## C.7 PROOF OF LEM. 4: SUB-GAUSSIAN MATRIX FREEDMAN INEQUALITY

We first note the following two lemmas about the tail bounds and symmetrized moment generating functions (MGFs) for matrix valued random variables (see Apps. C.9 and C.10 respectively for the proofs of Lems. 6 and 7).

**Lemma 6 (Sub-Gaussian matrix tail bounds)** *Let* $(\mathbf{X}_k \in \mathbb{R}^{m \times m})_{k \geq 1}$ *be a sequence of self-adjoint matrices adapted to a filtration* $\mathcal{F}_k$, *and let* $(\mathbf{A}_k \in \mathbb{R}^{m \times m})_{k \geq 1}$ *be a sequence of deterministic self-adjoint matrices. Define the variance parameter* $\sigma^2 \triangleq \|\sum_k \mathbf{A}_k\|_{\text{op}}$. *If, for a Rademacher random variable* $\varepsilon$ *independent of* $(\mathbf{X}_k, \mathcal{F}_k)_{k \geq 1}$, *we have*

$$
\log \mathbb{E}\big[\exp(2\varepsilon\theta\mathbf{X}_k)|\mathcal{F}_{k-1}\big] \preceq 2\theta^2 \mathbf{A}_k \quad \text{for all} \quad \theta \in \mathbb{R},
\tag{35}
$$

*then we also have*

$$
\mathbb{P}\Big[\lambda_{\max}\big(\sum_k \mathbf{X}_k\big) \geq t\Big] \leq m e^{-t^2/(8\sigma^2)} \quad \text{for all} \quad t \geq 0.
$$

**Lemma 7 (Symmetrized sub-Gaussian matrix MGF)** *For a fixed scalar $R$, let $\mathbf{X}$ be a self-adjoint matrix satisfying*

$$
\mathbb{E}\mathbf{X} = 0 \quad \text{and} \quad \mathbb{E}\mathbf{X}^q \preceq \big(\tfrac{q}{2}\big)! R^q \mathbf{I} \quad \text{for} \quad q \in 2\mathbb{N}.
\tag{36}
$$

*If $\varepsilon$ is a Rademacher random variable independent of $\mathbf{X}$, then*

$$
\mathbb{E}\exp(2\varepsilon\theta\mathbf{X}) \preceq \exp\big(2\theta^2 R^2 \mathbf{I}\big) \quad \text{for all} \quad \theta \in \mathbb{R}.
$$

The assumed conditions (25) allow us to apply Lem. 7 conditional on $(\mathbf{Y}_i)_{i<k}$ along with the operator monotonicity of $\log$ to find that

$$\log \mathbb{E}\Big[\exp(\varepsilon\theta\mathbf{Y}_k)|\{\mathbf{Y}_i\}_{i<k}\Big] \preceq 2\theta^2 R_k^2 \mathbf{I} \quad \text{for all} \quad \theta \in \mathbb{R},$$

for a Rademacher random variable $\varepsilon$ independent of $(\mathbf{Y}_k)_{k\geq 1}$. Moreover, $\left\|\sum_k \mathbf{A}_k\right\|_{\mathrm{op}} = \left\|\sum_k R_k^2 \mathbf{I}\right\|_{\mathrm{op}} = \sum_k R_k^2 = \sigma^2$. Thus, applying Lem. 6, we find that

$$\mathbb{P}[\lambda_{\max}(\textstyle\sum_i \mathbf{Y}_i) \geq t] \leq me^{-t^2/(8\sigma^2)} \text{ for all } t \geq 0.$$

As an immediate consequence, we also find that

$$\mathbb{P}[\lambda_{\max}(\textstyle\sum_i \mathbf{Y}_i) \geq \sqrt{8\sigma^2(t + \log m)}] \leq e^{-t} \quad \text{for all} \quad t \geq 0,$$

as claimed.

## C.8   PROOF OF LEM. 5: TAIL BOUNDS IMPLY MOMENT BOUNDS

We begin by bounding the moments of the shifted random variable $X = Z - a$. Note that $Z \geq 0$, so that $X \geq -a$. Next, note that $X = X_+ - X_-$ where $X_\pm = \max(\pm X, 0)$ and that $|X|^q = X_+^q + X_-^q$. Furthermore, $X_-^q \leq a^q$ by the nonnegativity of $Z$, so that $|X|^q \leq a^q + X_+^q$. For any $u > 0$, since $\mathbb{P}[X_+ > u] = \mathbb{P}[X > u] = \mathbb{P}[Z > a + u]$ for any $u > 0$, we apply the tail bounds on $Z$ to control the moments of $X_+$. In particular, we have

$$
\begin{aligned}
\mathbb{E}\big[X_+^q\big] &\overset{(i)}{=} q\int_0^\infty u^{q-1}\mathbb{P}[X_+ > u]du \\
&\overset{(ii)}{=} q\int_0^\infty (v\sqrt{t})^{q-1}\mathbb{P}[X_+ > v\sqrt{t}] \cdot \frac{v}{2\sqrt{t}}dt \\
&\overset{(iii)}{\leq} qv^q \int_0^\infty t^{q/2-1}e^{-t}dt \overset{(iv)}{=} qv^q\Gamma(\tfrac{q}{2}),
\end{aligned}
$$

where we have applied $(i)$ integration by parts, $(ii)$ the substitution $u = v\sqrt{t}$, and $(iii)$ the assumed tail bound for $Z$.

Since $Z = X + a$, the convexity of the function $t \mapsto t^q$ for $q \geq 1$, and Jensen's inequality imply that for each $q \in 2\mathbb{N}$, we have

$$
\begin{aligned}
\mathbb{E}Z^q \leq 2^{q-1}(a^q + \mathbb{E}|X|^q) \leq 2^{q-1}(2a^q + \mathbb{E}X_+^q) &\leq (2a)^q + 2^{q-1}qv^q\Gamma(\tfrac{q}{2}) \\
&= (2a)^q + 2^{q-1}qv^q(\tfrac{q}{2}-1)! \\
&\leq (2a + 2v)^q(\tfrac{q}{2})!
\end{aligned}
$$

where the last step follows since $x^q + y^q \leq (x + y)^q$ for all $q \in \mathbb{N}$ and $x, y \geq 0$. The proof is now complete.

## C.9   PROOF OF LEM. 6: SUB-GAUSSIAN MATRIX TAIL BOUNDS

The proof of this result is identical to that of Tropp (2012, Proof of Thm. 7.1) as the same steps are justified under our weaker assumption (35). Specifically, applying the arguments from Tropp (2012, Proof of Thm. 7.1), we find that

$$
\begin{aligned}
\mathbb{E}\big[\mathrm{tr}\exp(\textstyle\sum_{k=1}^n \theta\mathbf{X}_k)\big] &\leq \mathbb{E}\Big[\mathrm{tr}\exp\Big(\textstyle\sum_{k=1}^{n-1} \theta\mathbf{X}_k + \log\mathbb{E}\big[\exp(2\varepsilon\theta\mathbf{X}_n)|\mathcal{F}_{n-1}\big]\Big)\Big] \\
&\overset{(35)}{\leq} \mathbb{E}\Big[\mathrm{tr}\exp\Big(\textstyle\sum_{k=1}^{n-1} \theta\mathbf{X}_k + 2\theta^2\mathbf{A}_n\Big)\Big] \\
&\overset{(i)}{\leq} \mathrm{tr}\exp\big(2\theta^2\textstyle\sum_{k=1}^n \mathbf{A}_k\big) \overset{(ii)}{\leq} m\exp\big(2\theta^2\sigma^2\big), \quad\quad\quad (37)
\end{aligned}
$$

where step (i) follows by iterating the arguments over $k = n-1, \ldots, 1$ and step (ii) from the standard fact that $\mathrm{tr}(\exp(\mathbf{A})) \leq m\big\|\exp(\mathbf{A})\big\|_{\mathrm{op}} = m\exp(\|\mathbf{A}\|_{\mathrm{op}})$ for an $m \times m$ self-adjoint matrix $\mathbf{A}$. Next, applying the matrix Laplace transform method Tropp (2012, Prop. 3.1), for all $t > 0$, we have

$$
\begin{aligned}
\mathbb{P}\Big[\lambda_{\max}\big(\textstyle\sum_k \mathbf{X}_k\big) \geq t\big)\Big] &\leq \inf_{\theta>0}\Big\{e^{-\theta t} \cdot \mathbb{E}\big[\mathrm{tr}\exp(\textstyle\sum_{k=1}^n \theta\mathbf{X}_k)\big]\Big\} \\
&\overset{(37)}{\leq} m\inf_{\theta>0}\Big\{e^{-\theta t} \cdot e^{2\theta^2\sigma^2}\Big\} = me^{-t^2/(8\sigma^2)},
\end{aligned}
$$

where the last step follows from the choice $\theta = \frac{t}{4\sigma^2}$. The proof is now complete.

We have

$$\mathbb{E}[\exp(2\varepsilon\theta\mathbf{X})] = \mathbf{I} + \sum_{q=1}^{\infty} \frac{2^q\theta^q}{q!}\mathbb{E}[\varepsilon^q\mathbf{X}^q] \overset{(i)}{=} \mathbf{I} + \sum_{k=1}^{\infty} \frac{2^{2k}\theta^{2k}}{(2k)!}\mathbb{E}[\mathbf{X}^{2k}]$$

$$\overset{(ii)}{\preceq} \mathbf{I} + \sum_{k=1}^{\infty} \frac{2^{2k}\theta^{2k}\,k!R^{2k}}{(2k)!}\mathbf{I}$$

$$\overset{(iii)}{\preceq} \mathbf{I} + \sum_{k=1}^{\infty} \frac{(2\theta^2 R^2)^k}{k!}\mathbf{I} = \exp(2\theta^2 R^2\mathbf{I}),$$

where step (i) uses the facts that (a) $\mathbb{E}[\varepsilon^q] = \mathbf{1}(q \in 2\mathbb{N})$ and (b) $\mathbb{E}[\varepsilon^q\mathbf{X}^q] = \mathbb{E}[\varepsilon^q]\mathbb{E}[\mathbf{X}^q]$ since $\varepsilon$ is independent of $\mathbf{X}$, step (ii) follows from the assumed condition (36), and step (iii) from the fact that $\frac{2^k k!}{(2k)!} \leq \frac{1}{k!}$ (Boucheron et al., 2013, Proof of Thm. 2.1).

# D   PROOF OF THM. 3: RUNTIME AND INTEGRATION ERROR OF COMPRESS++

First, the runtime bound (9) follows directly by adding the runtime of COMPRESS(HALVE, $\mathfrak{g}$) as given by (5) in Thm. 1 and the runtime of THIN.

Recalling the notation (14) and (15) from App. A and noting the definition of the point sequences $\mathcal{S}_\text{C}$ and $\mathcal{S}_{\text{C++}}$ in Alg. 2, we obtain the following relationship between the different discrepancy vectors:

$$\phi_\text{C}(\mathcal{S}_\text{in}) = \frac{1}{n}\sum_{x\in\mathcal{S}_\text{in}}\delta_x - \frac{1}{2^{\mathfrak{g}}\sqrt{n}}\sum_{x\in\mathcal{S}_\text{C}}\delta_x,$$

$$\phi_\text{T}(\mathcal{S}_\text{C}) = \frac{1}{2^{\mathfrak{g}}\sqrt{n}}\sum_{x\in\mathcal{S}_\text{C}}\delta_x - \frac{1}{\sqrt{n}}\sum_{x\in\mathcal{S}_{\text{C++}}}\delta_x, \quad \text{and}$$

$$\phi_{\text{C++}}(\mathcal{S}_\text{in}) = \frac{1}{n}\sum_{x\in\mathcal{S}_\text{in}}\delta_x - \frac{1}{\sqrt{n}}\sum_{x\in\mathcal{S}_{\text{C++}}}\delta_x$$

$$= \phi_\text{C}(\mathcal{S}_\text{in}) + \phi_\text{T}(\mathcal{S}_\text{C}).$$

Noting the $\mathcal{G}^f$ property of HALVE and applying Thm. 1, we find that $\phi_\text{C}(\mathcal{S}_\text{in})(f)$ is sub-Gaussian with parameter $\nu_\text{C}(n)$ defined in (6). Furthermore, by assumption on THIN, given $\mathcal{S}_\text{C}$, the variable $\phi_\text{T}(\mathcal{S}_\text{C})(f)$ is $\nu_\text{C}(\frac{\ell_n}{2})$ sub-Gaussian. The claim now follows directly from Lem. 1.

# E   PROOF OF THM. 4: MMD GUARANTEES FOR COMPRESS++

Noting that MMD is a metric, and applying triangle inequality, we have

$$\text{MMD}_\mathbf{k}(\mathcal{S}_\text{in}, \mathcal{S}_{\text{C++}}) \leq \text{MMD}_\mathbf{k}(\mathcal{S}_\text{in}, \mathcal{S}_\text{C}) + \text{MMD}_\mathbf{k}(\mathcal{S}_\text{C}, \mathcal{S}_{\text{C++}}).$$

Since $\mathcal{S}_{\text{C++}}$ is the output of THIN($2^{\mathfrak{g}}$) with $\mathcal{S}_\text{C}$ as the input, applying the MMD tail bound assumption (38) with $|\mathcal{S}_\text{C}| = \sqrt{n}2^{\mathfrak{g}}$ substituted in place of $n$, we find that

$$\mathbb{P}\Big[\text{MMD}(\mathcal{S}_\text{C}, \mathcal{S}_{\text{C++}}) \geq a'_{2^{\mathfrak{g}}\sqrt{n}} + v'_{2^{\mathfrak{g}}\sqrt{n}}\sqrt{t}\Big] \leq e^{-t} \quad \text{for all} \quad t \geq 0.$$

Recall that $\ell_n/2 = 2^{\mathfrak{g}}\sqrt{n}$. Next, we apply Thm. 2 with HALVE to conclude that

$$\mathbb{P}[\text{MMD}_\mathbf{k}(\mathcal{S}_\text{in}, \mathcal{S}_\text{C}) \geq \widetilde{a}_n + \widetilde{v}_n \cdot \sqrt{t}] \leq e^{-t} \quad \text{for all} \quad t \geq 0.$$

Thus, we have

$$\mathbb{P}\Big[\text{MMD}_\mathbf{k}(\mathcal{S}_\text{in}, \mathcal{S}_{\text{C++}}) \geq a'_{\ell_n/2} + \widetilde{a}_n + (v'_{\ell_n/2} + \widetilde{v}_n)\sqrt{t}\Big] \leq 2 \cdot e^{-t} \quad \text{for all} \quad t \geq 0,$$

which in turn implies that

$$\mathbb{P}\Big[\text{MMD}_\mathbf{k}(\mathcal{S}_\text{in}, \mathcal{S}_{\text{C++}}) \geq a'_{\ell_n/2} + \widetilde{a}_n + (v'_{\ell_n/2} + \widetilde{v}_n)\sqrt{\log 2} + (v'_{\ell_n/2} + \widetilde{v}_n)\sqrt{t}\Big] \leq e^{-t} \quad \text{for all} \quad t \geq 0,$$

thereby yielding the claimed result.

# F   PROOFS OF EXS. 3, 4, 5, AND 6

We begin by defining the notions of sub-Gaussianity and **k**-sub-Gaussianity on an event.

**Definition 5 (Sub-Gaussian on an event)** *We say that a random variable $G$ is* sub-Gaussian on an event $\mathcal{E}$ with parameter $\sigma$ *if*

$$\mathbb{E}[\mathbf{1}[\mathcal{E}] \cdot \exp(\lambda \cdot G)] \leq \exp(\tfrac{\lambda^2 \sigma^2}{2}) \quad \text{for all} \quad \lambda \in \mathbb{R}.$$

**Definition 6 (k-sub-Gaussian on an event)** *For a kernel* **k***, we call a thinning algorithm* ALG **k**-*sub-Gaussian on an event $\mathcal{E}$ with parameter $v$ and shift $a$ if*

$$\mathbb{P}[\mathcal{E}, \mathrm{MMD}_{\mathbf{k}}(\mathcal{S}_{\mathrm{in}}, \mathcal{S}_{\mathrm{ALG}}) \geq a_n + v_n \sqrt{t} \mid \mathcal{S}_{\mathrm{in}}] \leq e^{-t} \quad \text{for all} \quad t \geq 0. \tag{38}$$

We will also make regular use of the unrolled representation (17) for the COMPRESS measure discrepancy $\psi_{\mathrm{C}}(\mathcal{S}_{\mathrm{in}})$ in terms of the HALVE inputs $(\mathcal{S}_{k,j}^{\mathrm{in}})_{j \in [4^k]}$ of size

$$n_k = 2^{\mathfrak{g}+1-k}\sqrt{n} \quad \text{for} \quad 0 \leq k \leq \beta_n. \tag{39}$$

For brevity, we will use the shorthand $\psi_{\mathrm{C}} \triangleq \psi_{\mathrm{C}}(\mathcal{S}_{\mathrm{in}})$, $\psi_{k,j}^{\mathrm{H}} \triangleq \psi_{\mathrm{H}}(\mathcal{S}_{k,j}^{\mathrm{in}})$, and $\psi_{\mathrm{T}} \triangleq \psi_{\mathrm{T}}(\mathcal{S}_{\mathrm{C}})$ hereafter.

## F.1   PROOF OF EX. 3: KT-SPLIT-COMPRESS

For HALVE $=$ KT-SPLIT$(\frac{\ell^2}{n4^{\mathfrak{g}+1}(\beta_n+1)}\delta)$ when applied to an input of size $\ell$, the proof of Thm. 1 in Dwivedi & Mackey (2022) identifies a sequence of events $\mathcal{E}_{k,j}$ and random signed measures $\tilde{\psi}_{k,j}$ such that, for each $0 \leq k \leq \beta_n$, $j \in [4^k]$, and $f$ with $\|f\|_{\mathbf{k}} = 1$,

(a) $\mathbb{P}[\mathcal{E}_{k,j}^c] \overset{(i)}{\leq} \frac{n_k^2}{n4^{\mathfrak{g}+1}(\beta_n+1)}\frac{\delta}{2} \overset{(ii)}{=} \frac{1}{2}\frac{\delta}{4^k(\beta_n+1)}$,

(b) $\mathbf{1}[\mathcal{E}_{k,j}]\psi_{k,j}^{\mathrm{H}} = \mathbf{1}[\mathcal{E}_{k,j}]\tilde{\psi}_{k,j}$, and

(c) $\tilde{\psi}_{k,j}(f)$ is $n_k \, \nu_{\mathrm{H}}(n_k)$ sub-Gaussian (7) given $(\tilde{\psi}_{k',j'})_{k'>k,j'\geq 1}$ and $(\tilde{\psi}_{k,j'})_{j'<j}$,

where step (ii) follows from substituting the definition $n_k = 2^{\mathfrak{g}+1-k}\sqrt{n}$ (39). To establish step (ii) in property (a), we use the definition[1] of KT-SPLIT$(\frac{n_k^2}{n4^{\mathfrak{g}+1}(\beta_n+1)}\delta)$ for an input of size $n_k$, which implies that $\delta_i = \frac{n_k}{n4^{\mathfrak{g}+1}(\beta_n+1)}\delta$ in the notation of Dwivedi & Mackey (2022). The proof of Thm. 1 in Dwivedi & Mackey (2022) then implies that

$$\mathbb{P}[\mathcal{E}_{k,j}^c] \leq \sum_{i=1}^{n_k/2} \delta_i = \frac{n_k}{2}\frac{n_k}{n4^{\mathfrak{g}+1}(\beta_n+1)}\delta = \frac{n_k^2}{n4^{\mathfrak{g}+1}(\beta_n+1)}\frac{\delta}{2}.$$

Hence, on the event $\mathcal{E} = \bigcap_{k,j}\mathcal{E}_{k,j}$, these properties hold simultaneously for all HALVE calls made by COMPRESS, and, by the union bound,

$$\mathbb{P}[\mathcal{E}^c] \leq \sum_{k=0}^{\beta_n}\sum_{j=1}^{4^k}\mathbb{P}[\mathcal{E}_{k,j}^c] \leq \sum_{k=0}^{\beta_n}4^k\frac{1}{2}\frac{\delta}{4^k(\beta_n+1)} = \frac{\delta}{2}. \tag{40}$$

Now fix any $f$ with $\|f\|_{\mathbf{k}} = 1$. We invoke the measure discrepancy representation (17), the equivalence of $\psi_{k,j}^{\mathrm{H}}$ and $\tilde{\psi}_{k,j}$ on $\mathcal{E}$, the nonnegativity of the exponential, and Lem. 1 in turn to find

$$
\begin{aligned}
\mathbb{E}[\mathbf{1}[\mathcal{E}] \cdot \exp(\lambda \cdot \phi_{\mathrm{C}}(f))] &= \mathbb{E}[\mathbf{1}[\mathcal{E}] \cdot \exp(\lambda \cdot \tfrac{1}{n}\psi_{\mathrm{C}}(f))] \\
&= \mathbb{E}[\mathbf{1}[\mathcal{E}] \cdot \exp(\lambda \cdot \tfrac{1}{n}\sqrt{n}2^{-\mathfrak{g}-1}\textstyle\sum_{k=0}^{\beta_n}\sum_{j=1}^{4^k}2^{-k}\psi_{k,j}^{\mathrm{H}}(f))] \\
&= \mathbb{E}[\mathbf{1}[\mathcal{E}] \cdot \exp(\lambda \cdot \tfrac{1}{n}\sqrt{n}2^{-\mathfrak{g}-1}\textstyle\sum_{k=0}^{\beta_n}\sum_{j=1}^{4^k}2^{-k}\tilde{\psi}_{k,j}(f))] \\
&\leq \mathbb{E}[\exp(\lambda \cdot \tfrac{1}{n}\sqrt{n}2^{-\mathfrak{g}-1}\textstyle\sum_{k=0}^{\beta_n}\sum_{j=1}^{4^k}2^{-k}\tilde{\psi}_{k,j}(f))] \\
&\leq \exp(\tfrac{\lambda^2 \nu_{\mathrm{C}}^2(n)}{2}) \quad \text{for} \quad \nu_{\mathrm{C}}^2(n) = \textstyle\sum_{k=0}^{\beta_n}4^{-k}\nu_{\mathrm{H}}^2(n_k)
\end{aligned}
$$

so that $\phi_{\mathrm{C}}(f)$ is $\nu_{\mathrm{C}}$ sub-Gaussian on $\mathcal{E}$.

## F.2 PROOF OF EX. 4: KT-COMPRESS

For HALVE = symmetrized $\mathrm{KT}(\frac{\ell^2}{n4^{\mathfrak{g}+1}(\beta_n+1)}\delta)$ when applied to an input of size $\ell$, the proofs of Thms. 1–4 in Dwivedi & Mackey (2022) identify a sequence of events $\mathcal{E}_{k,j}$ and random signed measures $\tilde{\psi}_{k,j}$ such that, for each $0 \le k \le \beta_n$ and $j \in [4^k]$,

(a) $\mathbb{P}[\mathcal{E}_{k,j}^c] \le \frac{1}{2}\frac{\delta}{4^k(\beta_n+1)}$,

(b) $\mathbf{1}[\mathcal{E}_{k,j}]\psi_{k,j}^{\mathrm{H}} = \mathbf{1}[\mathcal{E}_{k,j}]\tilde{\psi}_{k,j}$,

(c) $\mathbb{P}[\frac{1}{n_k}\|\tilde{\psi}_{k,j}(\mathbf{k})\|_{\mathbf{k}} \ge a_{n_k} + v_{n_k}\sqrt{t} \mid (\tilde{\psi}_{k',j'})_{k'>k,j'\ge 1}, (\tilde{\psi}_{k,j'})_{j'<j}] \le e^{-t}$ for all $t \ge 0$, and

(d) $\mathbb{E}[\tilde{\psi}_{k,j}(\mathbf{k}) \mid (\tilde{\psi}_{k',j'})_{k'>k,j'\ge 1}, (\tilde{\psi}_{k,j'})_{j'<j}] = 0$,

where $n_k = 2^{\mathfrak{g}+1-k}\sqrt{n}$ was defined in (39). We derive property (a) exactly as in App. F.1.

Hence, on the event $\mathcal{E} = \bigcap_{k,j}\mathcal{E}_{k,j}$, these properties hold simultaneously for all HALVE calls made by COMPRESS, and, by the union bound (40), $\mathbb{P}[\mathcal{E}^c] \le \frac{\delta}{2}$.

Furthermore, we may invoke the measure discrepancy representation (17), the equivalence of $\psi_{k,j}^{\mathrm{H}}$ and $\tilde{\psi}_{k,j}$ on $\mathcal{E}$, the nonnegativity of the exponential, and the proof of Thm. 2 in turn to find

$$\mathbb{P}[\mathcal{E}, \mathrm{MMD}(\mathcal{S}_{\mathrm{in}}, \mathcal{S}_{\mathrm{C}}) \ge \tilde{a}_n + \tilde{v}_n\sqrt{t} \mid \mathcal{S}_{\mathrm{in}}] = \mathbb{P}[\mathcal{E}, \tfrac{1}{n}\|\psi_{\mathrm{C}}(\mathbf{k})\|_{\mathbf{k}} \ge \tilde{a}_n + \tilde{v}_n\sqrt{t} \mid \mathcal{S}_{\mathrm{in}}]$$

$$= \mathbb{P}[\mathcal{E}, \tfrac{1}{n}\|\sqrt{n}2^{-\mathfrak{g}-1}\textstyle\sum_{k=0}^{\beta_n}\sum_{j=1}^{4^k}2^{-k}\tilde{\psi}_{k,j}(\mathbf{k})\|_{\mathbf{k}} \ge \tilde{a}_n + \tilde{v}_n\sqrt{t} \mid \mathcal{S}_{\mathrm{in}}]$$

$$\le \mathbb{P}[\tfrac{1}{n}\|\sqrt{n}2^{-\mathfrak{g}-1}\textstyle\sum_{k=0}^{\beta_n}\sum_{j=1}^{4^k}2^{-k}\tilde{\psi}_{k,j}(\mathbf{k})\|_{\mathbf{k}} \ge \tilde{a}_n + \tilde{v}_n\sqrt{t} \mid \mathcal{S}_{\mathrm{in}}] \le e^{-t} \quad \text{for all} \quad t \ge 0,$$

so that COMPRESS is $\mathbf{k}$-sub-Gaussian on $\mathcal{E}$ with parameters $(\tilde{v}, \tilde{a})$.

## F.3 PROOF OF EX. 5: KT-SPLIT-COMPRESS++

For THIN = KT-SPLIT$(\frac{\mathfrak{g}}{\mathfrak{g}+2^{\mathfrak{g}}(\beta_n+1)}\delta)$ and HALVE = KT-SPLIT$(\frac{\ell^2}{4n2^{\mathfrak{g}}(\mathfrak{g}+2^{\mathfrak{g}}(\beta_n+1))}\delta)$ when applied to an input of size $\ell$, the proof of Thm. 1 in Dwivedi & Mackey (2022) identifies a sequence of events $\mathcal{E}_{k,j}$ and $\mathcal{E}_{\mathrm{T}}$ and random signed measures $\tilde{\psi}_{k,j}$ and $\tilde{\psi}_{\mathrm{T}}$ such that, for each $0 \le k \le \beta_n$, $j \in [4^k]$, and $f$ with $\|f\|_{\mathbf{k}} = 1$,

(a) $\mathbb{P}[\mathcal{E}_{k,j}^c] \overset{(i)}{\le} \frac{n_k^2}{4n2^{\mathfrak{g}}(\mathfrak{g}+2^{\mathfrak{g}}(\beta_n+1))}\frac{\delta}{2} \overset{(ii)}{=} \frac{2^{\mathfrak{g}}}{4^k(\mathfrak{g}+2^{\mathfrak{g}}(\beta_n+1))}\frac{\delta}{2}$ and $\mathbb{P}[\mathcal{E}_{\mathrm{T}}^c] \overset{(iii)}{\le} \frac{\mathfrak{g}}{\mathfrak{g}+2^{\mathfrak{g}}(\beta_n+1)}\frac{\delta}{2}$,

(b) $\mathbf{1}[\mathcal{E}_{k,j}]\psi_{k,j}^{\mathrm{H}} = \mathbf{1}[\mathcal{E}_{k,j}]\tilde{\psi}_{k,j}$ and $\mathbf{1}[\mathcal{E}_{\mathrm{T}}]\psi_{\mathrm{T}} = \mathbf{1}[\mathcal{E}_{\mathrm{T}}]\tilde{\psi}_{\mathrm{T}}$, and

(c) $\tilde{\psi}_{k,j}(f)$ is $n_k\,\nu_{\mathrm{H}}(n_k)$ sub-Gaussian (12) given $(\tilde{\psi}_{k',j'})_{k'>k,j'\ge 1}$ and $(\tilde{\psi}_{k,j'})_{j'<j}$ and $\tilde{\psi}_{\mathrm{T}}$ is $\frac{\ell_n}{2}\,\nu_{\mathrm{T}}(\frac{\ell_n}{2})$ sub-Gaussian (12) given $\mathcal{S}_{\mathrm{C}}$.

Here, step (i) and (ii) follow exactly as in steps (i) and (ii) of property (a) in App. F.1. For step (iii), we use the definition[1] of KT-SPLIT$(\frac{\mathfrak{g}}{\mathfrak{g}+2^{\mathfrak{g}}(\beta_n+1)}\delta)$ for an input of size $2^{\mathfrak{g}}\sqrt{n}$, which implies that $\delta_i = \frac{\mathfrak{g}}{\sqrt{n}2^{\mathfrak{g}}(\mathfrak{g}+2^{\mathfrak{g}}(\beta_n+1))}\delta$ in the notation of Dwivedi & Mackey (2022). The proof of Thm. 1 in Dwivedi & Mackey (2022) then implies that

$$\mathbb{P}[\mathcal{E}_{\mathrm{T}}^c] \le \textstyle\sum_{j=1}^{\mathfrak{g}}\frac{2^{j-1}}{\mathfrak{g}}\sum_{i=1}^{2^{\mathfrak{g}-j}\sqrt{n}}\delta_i = \sum_{j=1}^{\mathfrak{g}}\frac{2^{j-1}}{\mathfrak{g}}2^{\mathfrak{g}-j}\sqrt{n}\frac{1}{\sqrt{n}2^{\mathfrak{g}}}\cdot\frac{\mathfrak{g}}{\mathfrak{g}+2^{\mathfrak{g}}(\beta_n+1)}\delta = \frac{\mathfrak{g}}{\mathfrak{g}+2^{\mathfrak{g}}(\beta_n+1)}\frac{\delta}{2},$$

as claimed.

Hence, on the event $\mathcal{E} = \bigcap_{k,j}\mathcal{E}_{k,j} \cap \mathcal{E}_{\mathrm{T}}$, these properties hold simultaneously for all HALVE calls made by COMPRESS, and, repeating an argument similar to the union bound (40),

$$\mathbb{P}[\mathcal{E}^c] \le \mathbb{P}[\mathcal{E}_{\mathrm{T}}^c] + \textstyle\sum_{k=0}^{\beta_n}\sum_{j=1}^{4^k}\mathbb{P}[\mathcal{E}_{k,j}^c] \le \frac{\mathfrak{g}}{\mathfrak{g}+2^{\mathfrak{g}}(\beta_n+1)}\frac{\delta}{2} + \sum_{k=0}^{\beta_n}4^k\frac{2^{\mathfrak{g}}}{4^k(\mathfrak{g}+2^{\mathfrak{g}}(\beta_n+1))}\frac{\delta}{2} = \frac{\delta}{2}. \quad (41)$$

Moreover, since $\phi_{\mathrm{C}++} = \frac{1}{n}(\psi_{\mathrm{C}} + \psi_{\mathrm{T}})$, Lem. 1 and the argument of App. F.1 together imply that $\phi_{\mathrm{C}}(f)$ is $\nu_{\mathrm{C}++}$ sub-Gaussian on $\mathcal{E}$ for each $f$ with $\|f\|_{\mathbf{k}} = 1$.

In the notation of Ex. 2, define

$$\frac{\ell_n}{2} a_{\ell_n} = \sqrt{n} a'_{\ell_n/2} = C_a \sqrt{\|\mathbf{k}\|_\infty}, \quad \text{and}$$

$$\frac{\ell_n}{2} v_{\ell_n} = \sqrt{n} v'_{\ell_n/2} = C_v \sqrt{\|\mathbf{k}\|_\infty \log\left(\frac{6(n-\sqrt{n}(2^{\mathfrak{g}}-\mathfrak{g}))}{\delta}\right)} \, \mathfrak{M}_{\mathcal{S}_{\mathrm{in}},\mathbf{k}}.$$

Since HALVE $=$ symmetrized $\mathrm{KT}\left(\frac{\ell^2}{4n2^{\mathfrak{g}}(\mathfrak{g}+2^{\mathfrak{g}}(\beta_n+1))}\delta\right)$ for inputs of size $\ell$ and THIN $=$ $\mathrm{KT}\left(\frac{\mathfrak{g}}{\mathfrak{g}+2^{\mathfrak{g}}(\beta_n+1)}\delta\right)$, the proofs of Thms. 1–4 in Dwivedi & Mackey (2022) identify a sequence of events $\mathcal{E}_{k,j}$ and $\mathcal{E}_{\mathrm{T}}$ and random signed measures $\tilde{\psi}_{k,j}$ and $\tilde{\psi}_{\mathrm{T}}$ such that, for each $0 \le k \le \beta_n$ and $j \in [4^k]$,

(a) $\mathbb{P}[\mathcal{E}_{k,j}^c] \le \frac{2^{\mathfrak{g}}}{4^k(\mathfrak{g}+2^{\mathfrak{g}}(\beta_n+1))}\frac{\delta}{2}$ and $\mathbb{P}[\mathcal{E}_{\mathrm{T}}^c] \le \frac{\mathfrak{g}}{\mathfrak{g}+2^{\mathfrak{g}}(\beta_n+1)}\frac{\delta}{2}$,

(b) $\mathbf{1}[\mathcal{E}_{k,j}]\psi_{k,j}^{\mathrm{H}} = \mathbf{1}[\mathcal{E}_{k,j}]\tilde{\psi}_{k,j}$ and $\mathbf{1}[\mathcal{E}_{\mathrm{T}}]\psi_{\mathrm{T}} = \mathbf{1}[\mathcal{E}_{\mathrm{T}}]\tilde{\psi}_{\mathrm{T}}$,

(c) $\mathbb{P}[\frac{1}{n_k}\|\tilde{\psi}_{k,j}(\mathbf{k})\|_{\mathbf{k}} \ge a_{n_k} + v_{n_k}\sqrt{t} \mid (\tilde{\psi}_{k',j'})_{k'>k,j'\ge 1}, (\tilde{\psi}_{k,j'})_{j'<j}] \le e^{-t}$ and $\mathbb{P}[\frac{2}{\ell_n}\|\tilde{\psi}_{\mathrm{T}}(\mathbf{k})\|_{\mathbf{k}} \ge a'_{\ell_n/2} + v'_{\ell_n/2}\sqrt{t} \mid \mathcal{S}_{\mathrm{C}}] \le e^{-t}$ for all $t \ge 0$, and

(d) $\mathbb{E}[\tilde{\psi}_{k,j}(\mathbf{k}) \mid (\tilde{\psi}_{k',j'})_{k'>k,j'\ge 1}, (\tilde{\psi}_{k,j'})_{j'<j}] = 0$.

We derive property (a) exactly as in App. F.3. Hence, on the event $\mathcal{E} = \bigcap_{k,j}\mathcal{E}_{k,j} \cap \mathcal{E}_{\mathrm{T}}$, these properties hold simultaneously for all HALVE calls made by COMPRESS and

$$\widetilde{\zeta}_{\mathrm{H}}(\ell_n) = \widetilde{\zeta}_{\mathrm{T}}(\tfrac{\ell_n}{2}) = C_v \sqrt{\|\mathbf{k}\|_\infty \log\left(\frac{6(n-\sqrt{n}(2^{\mathfrak{g}}-\mathfrak{g}))}{\delta}\right)} \, \mathfrak{M}_{\mathcal{S}_{\mathrm{in}},\mathbf{k}}.$$

Moreover, by the union bound (41), $\mathbb{P}[\mathcal{E}^c] \le \frac{\delta}{2}$.

Finally, since $\phi_{\mathrm{C++}} = \frac{1}{n}(\psi_{\mathrm{C}} + \psi_{\mathrm{T}})$ and the argument of App. F.2 implies that COMPRESS is $\mathbf{k}$-sub-Gaussian on $\mathcal{E}$ with parameters $(\tilde{v}, \tilde{a})$, the triangle inequality implies that COMPRESS++ is $\mathbf{k}$-sub-Gaussian on $\mathcal{E}$ with parameters $(\hat{v}, \hat{a})$ as in App. E.

# G SUPPLEMENTARY DETAILS FOR EXPERIMENTS

In this section, we provide supplementary experiment details deferred from Sec. 5, as well as some additional results.

In the legend of each MMD plot, we display an empirical rate of decay. In all experiments involving kernel thinning, we set the algorithm failure probability parameter $\delta = \frac{1}{2}$ and compare $\mathrm{KT}(\delta)$ to COMPRESS and COMPRESS++ with HALVE and THIN set as in Exs. 4 and 6 respectively.

## G.1 MIXTURE OF GAUSSIAN TARGET DETAILS AND MMD PLOTS

For the target used for coreset visualization in Fig. 3, the mean locations are on two concentric circles of radii 10 and 20, and are given by

$$\mu_j = \alpha_j \begin{bmatrix} \sin(j) \\ \cos(j) \end{bmatrix} \quad \text{where } \alpha_j = 10 \cdot \mathbf{1}(j \le 16) + 20 \cdot \mathbf{1}(j > 16) \qquad \text{for } j = 1, 2, \ldots, 32.$$

Here we also provide additional results with mixture of Gaussian targets given by $\mathbb{P} = \frac{1}{M}\sum_{j=1}^{M}\mathcal{N}(\mu_j, \mathbf{I}_d)$ for $M \in \{4, 6, 8\}$. The mean locations for these are given by

$$\mu_1 = [-3, 3]^\top, \quad \mu_2 = [-3, 3]^\top, \quad \mu_3 = [-3, -3]^\top, \quad \mu_4 = [3, -3]^\top,$$
$$\mu_5 = [0, 6]^\top, \qquad \mu_6 = [-6, 0]^\top, \quad \mu_7 = [6, 0]^\top, \qquad \mu_8 = [0, -6]^\top.$$

Fig. 4 plots the MMD errors of KT and herding experiments for the mixture of Gaussians targets with 4, 6 and 8 centers, and notice again that COMPRESS++ provides a competitive performance to the original algorithm, in fact suprisingly, improves upon herding.

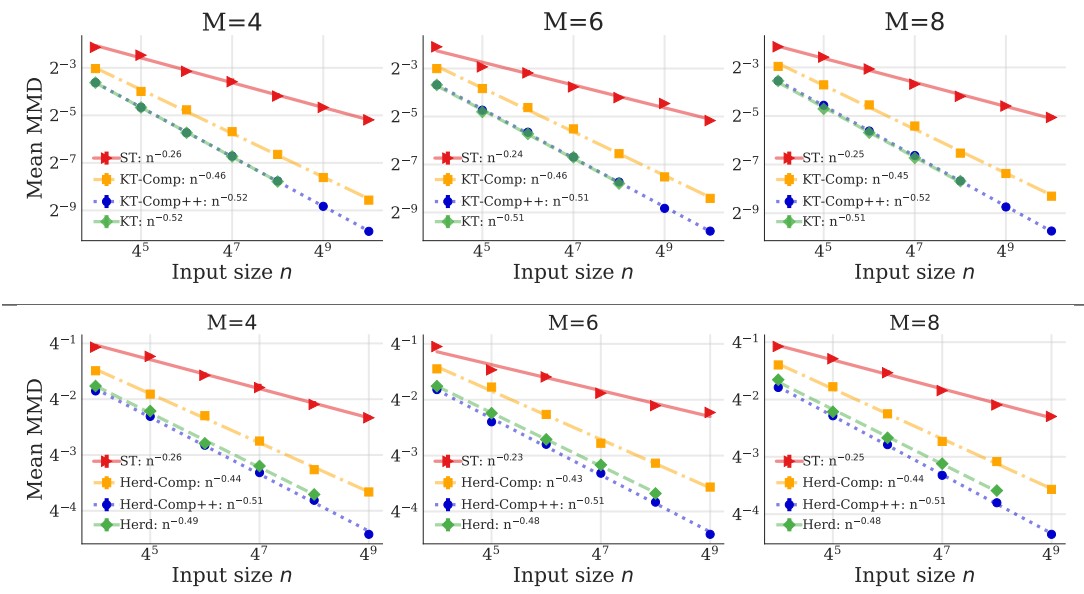

**Figure 4:** For $M$-component mixture of Gaussian targets, KT-COMPRESS++ and Herd-COMPRESS++ improve upon the MMD of i.i.d. sampling (ST) and closely track or improve upon the error of their quadratic-time input algorithms, KT and kernel herding (Herd). See App. G.1 for more details.

## G.2   DETAILS OF MCMC TARGETS

Our set-up for the MCMC experiments is identical to that of Dwivedi & Mackey (2021, Sec. 6), except that we use all post-burn-in points to generate our Goodwin and Lotka-Volterra input point sequences $\mathcal{S}_{\text{in}}$ instead of only the odd indices. In particular, we use the MCMC output of Riabiz et al. (2020b) described in (Riabiz et al., 2020a, Sec. 4) and perform thinning experiments after discarding the burn-in points. To generate an input $\mathcal{S}_{\text{in}}$ of size $n$ for a thinning algorithm, we downsample the post-burn-in points using standard thinning. For Hinch, we additionally do coordinate-wise normalization by subtracting the sample mean and dividing by sample standard deviation of the post-burn-in-points.

In Sec. 5, RW and ADA-RW respectively refer to Gaussian random walk and adaptive Gaussian random walk Metropolis algorithms (Haario et al., 1999) and MALA and pMALA respectively refer to the Metropolis-adjusted Langevin algorithm (Roberts & Tweedie, 1996) and pre-conditioned MALA (Girolami & Calderhead, 2011). For Hinch experiments, RW 1 and RW 2 refer to two independent runs of Gaussian random walk, and "Tempered" denotes the runs targeting a tempered Hinch posterior. For more details on the set-up, we refer the reader to Dwivedi & Mackey (2021, Sec. 6.3, App. J.2).

## H   STREAMING VERSION OF COMPRESS

COMPRESS can be efficiently implemented in a streaming fashion (Alg. 3) by viewing the recursive steps in Alg. 1 as different levels of processing, with the bottom level denoting the input points and the top level denoting the output points. The streaming variant of the algorithm efficiently maintains memory at several levels and processes inputs in batches of size $4^{\mathfrak{g}+1}$. At any level $i$ (with $i = 0$ denoting the level of the input points), whenever there are $2^i 4^{\mathfrak{g}+1}$ points, the algorithm runs HALVE on the points in this level, appends the output of size $2^{i-1} 4^{\mathfrak{g}+1}$ to the points at level $i+1$, and empties the memory at level $i$ (and thereby level $i$ never stores more than $2^i 4^{\mathfrak{g}+1}$ points). In this fashion, just after processing $n = 4^{k+\mathfrak{g}+1}$ points, the highest level is $k+1$, which contains a compressed coreset of size $2^{k-1} 4^{\mathfrak{g}+1} = 2^{k+\mathfrak{g}+1} 2^{\mathfrak{g}} = \sqrt{n} 2^{\mathfrak{g}}$ (outputted by running HALVE at level $k$ for the first time), which is the desired size for the output of COMPRESS.

**Algorithm 3:** COMPRESS (Streaming) – Outputs stream of coresets of size $2^{\mathfrak{g}}\sqrt{n}$ for $n = 4^{k+\mathfrak{g}+1}$ and $k \in \mathbb{N}$

---

**Input:** halving algorithm HALVE, oversampling parameter $\mathfrak{g}$, stream of input points $x_1, x_2, \ldots$

$\mathcal{S}_0 \leftarrow \{\}$                // Initialize empty level 0 coreset
**for** $t = 1, 2, \ldots,$ **do**
    $\mathcal{S}_0 \leftarrow \mathcal{S}_0 \cup (x_j)_{j=1+(t-1)\cdot 4^{\mathfrak{g}+1}}^{t\cdot 4^{\mathfrak{g}+1}}$      // Process input in batches of size $4^{\mathfrak{g}+1}$
    **if** $t == 4^j$ for $j \in \mathbb{N}$ **then**
        $\mathcal{S}_{j+1} \leftarrow \{\}$          // Initialize level $j + 1$ coreset after processing $4^{j+\mathfrak{g}+1}$ input points
    **end**
    **for** $i = 0, \ldots, \lceil \log_4 t \rceil + 1$ **do**
        **if** $|\mathcal{S}_i| == 2^i 4^{\mathfrak{g}+1}$ **then**
            $\mathcal{S} \leftarrow \text{HALVE}(\mathcal{S}_i)$      // Halve level $i$ coreset to size $2^{i-1}4^{\mathfrak{g}+1}$
            $\mathcal{S}_{i+1} \leftarrow \mathcal{S}_{i+1} \cup \mathcal{S}$      // Update level $i+1$ coreset: has size $\in \{1,2,3,4\} \cdot 2^{i-1}4^{\mathfrak{g}+1}$
            $\mathcal{S}_i \leftarrow \{\}$        // Empty coreset at level $i$
        **end**
    **end**
    **if** $t == 4^j$ for $j \in \mathbb{N}$ **then**
        **output** $\mathcal{S}_{j+1}$      // Coreset of size $\sqrt{n}2^{\mathfrak{g}}$ with $n \triangleq t4^{\mathfrak{g}+1}$ and $t = 4^j$ for $j \in \mathbb{N}$
    **end**
**end**

---

Our next result analyzes the space complexity of the streaming variant (Alg. 3) of COMPRESS. The intuition for gains in memory requirements is very similar to that for running time, as we now maintain (and run HALVE) on subsets of points with size much smaller than the input sequence. We count the number of data points stored as our measure of memory.

**Proposition 1 (COMPRESS Streaming Memory Bound)** *Let* HALVE *store* $s_H(n)$ *data points on inputs of size* $n$. *Then, after completing iteration* $t$, *the streaming implementation of* COMPRESS *(Alg. 3) has used at most* $s_C(t) = 4^{\mathfrak{g}+3}\sqrt{t} + s_H(2^{\mathfrak{g}+1}\sqrt{t})$ *data points of memory.*

**Proof** At time $t$, we would like to estimate the space usage of the algorithm. At the $i$th level of memory, we can have at most $2^{i+2}4^{\mathfrak{g}}$ data points. Since we are maintaining a data set of size at most $\sqrt{t}4^{\mathfrak{g}}$ at time $t$, there are at most $\frac{\log t}{2}$ levels. Thus, the maximum number of points stored at time $t$ is bounded by

$$\sum_{i=0}^{0.5 \log t} 2^{i+2}4^{\mathfrak{g}} \leq 4^{\mathfrak{g}+3}\sqrt{t}.$$

Furthermore, at any time up to time $t$, we have run HALVE on a point sequence of size at most $\sqrt{t}2^{\mathfrak{g}+1}$ which requires storing at most $s_H(\sqrt{t}2^{\mathfrak{g}+1})$ additional points. $\qquad\square$

**Example 7 (KT-COMPRESS and KT-COMPRESS++)** First consider the streaming variant of COMPRESS with HALVE = symmetrized $\text{KT}(\frac{\ell}{2n-\ell_n}\delta)$ for HALVE inputs of size $\ell$ as in Ex. 4. Since $s_{KT}(n) \leq n$ (Dwivedi & Mackey, 2021, Sec. 3), Prop. 1 implies that $s_C(n) \leq 4^{\mathfrak{g}+4}\sqrt{n}$.

Next consider COMPRESS++ with the streaming variant of COMPRESS, with HALVE = symmetrized $\text{KT}(\frac{\ell}{2n-\ell_n+2\mathfrak{g}\sqrt{n}}\delta)$ when applied to an input of size $\ell$, and THIN = $\text{KT}(\frac{\mathfrak{g}}{\sqrt{n}-2^{\mathfrak{g}}+\mathfrak{g}}\delta)$ as in Ex. 6. The space complexity $s_{C++}(n) = s_C(n) + s_{KT}(\ell_n) = 4^{\mathfrak{g}+4}\sqrt{n} + \ell_n \leq 4^{\mathfrak{g}+5}\sqrt{n}$. Setting $\mathfrak{g}$ as in Ex. 6, we get $s_{C++}(n) = \mathcal{O}(\sqrt{n}\log^2 n)$. ∎

