# OpenReview forum: "Distribution Compression in Near-Linear Time"
_ICLR.cc/2022/Conference — ICLR 2022 Poster_

### Official Review · Reviewer_hLMP · 2021-11-02

**Correctness:** 4
**Technical Novelty And Significance:** 2
**Empirical Novelty And Significance:** 2
**Recommendation:** 6
**Confidence:** 3

**Main Review:**

The paper introduces a new framework to improve on runtime of existing thinning procedures. The framework is quite simple and analysis is quite straight forward (except error analysis using Sub-Gamma) . Eventhough the framework is simple, it helps improve the runtime of existing thinning algorithms (from magnitude of days to hours). The runtime improvements are only pronounced for higher input size (10^3) (not sure if mentioned applications have such high sample size requirements).


**Summary Of The Paper:**

The paper introduces meta algorithms "Compress" and "Compress++" that take existing thinning algorithm as a subroutine and improves on their runtime while incurring marginally more error.

The algorithm "Compress" runs in recursive fashion: divides input set into four parts, runs Compress on each part independently, combines the resulting sets and halves the combined set using "HALVE" algorithm. As an example, the authors demonstrate that one can use KT-SPLIT as HALVE algorithm and get faster runtimes to compress albeit suffering extra error (upto log factors). Compress ensures that KT-SPLIT runs on sets of much smaller size.

 They build on Compress to obtain better error rates while sacrificing on runtimes (upto log factors). For this, they use the thinning algorithm on much smaller set obtained after running Compress. The runtimes are quite easy to derive based on recursion of algorithms. They use sub-gamma property to prove the error guarantees of the algorithms. They demonstrate the faster runtimes of their algorithms using high dimensional Monte Carlo samples.

**Summary Of The Review:**

Even though the framework and analysis is simple, I believe community can benefit from these faster thinning meta algorithms. I am not sure how well "distribution compression" fits ICLR.

---

> ### Author Response · Authors · 2021-11-22
> **Author Response**
>
> Thank you for the time you’ve taken to review our work and for your positive feedback!  We are delighted that you found our work on rapidly learning compressed representations to be of benefit to the community.  We strove to make the Compress algorithm as simple and low-overhead as possible to maximize its utility (after developing more complicated meta-algorithms and discarding them in favor of this one), so we are glad that you also appreciated the simplicity of our framework and analysis.

---

> > ### Author Response · Authors · 2021-11-22
> > **Significance and subtlety of error guarantees**
> >
> > In addition, we will clarify in the revision that while the runtime of compress is immediate upon unrolling the recursion, the error guarantee of Equation 4 is more subtle: here, Compress benefits significantly from random cancellations among the conditionally independent and mean-zero errors from Halve. Without these properties, the errors from each Halve call could compound without cancellation leading to a significant degradation in quality. Fortunately, as we note in Remark 4, we can convert any halving algorithm into one that satisfies the mean zero condition of Thm. 1 without impacting integration error, by symmetrization, i.e., by returning either the outputted half or its complement with equal probability.
> >
> > In the revision we will emphasize that this subtle point can have a substantial impact in practice.  For example, the kernel herding algorithm is completely deterministic and hence does not satisfy the mean zero condition of Thm. 1.  Therefore, our experiments in Figure 3 used a symmetrized version of Herding as the input halving algorithm to recover better-than-iid compression.  However, strikingly (and in line with the theoretical explanation above), if the same experiment is run with standard unsymmetrized kernel herding, the MMD error of Herd-Compress does not decay at all!  Please see the following figure which we will incorporate into the final revision: https://drive.google.com/drive/folders/1Q_hs-jWC2pqMk62Sh00hUHAR5WCPsGu-?usp=sharing

---

> > > ### Comment · Reviewer_hLMP · 2021-11-30
> > > **Thanks for response**
> > >
> > > That new figure can be beneficial

---

> > ### Author Response · Authors · 2021-11-22
> > **Are input sizes larger than 10^3 common?**
> >
> > The reviewer noted that gains were most pronounced for input sizes larger than 10^3 and wondered whether input sets were commonly this large.  Yes, input sizes larger than 10^3 are quite common, especially when the input points come from an MCMC chain which can require a long chain length simply to adequately explore the target distribution.  In fact, in the 12 posterior inference experiments of Sec. 4, the input chains (even after burn-in removal) all had length at least n = 10^6.

---

### Official Review · Reviewer_FqWv · 2021-11-03

**Correctness:** 3
**Technical Novelty And Significance:** 3
**Empirical Novelty And Significance:** 4
**Recommendation:** 8
**Confidence:** 3

**Main Review:**

I believe the paper makes a good contribution to the field by proposing a general reduction scheme that improves the runtime of existing algorithms at cost of minimal increase in error.
The underlying idea is clear and simple (in hindsight), which adds to the strength of the contribution. Thus I recommend accept. I have following suggestions/comments, and look forward to hearing back from authors.

## Major Remarks

1. I think the proof sketch of MMD guarantees should be added to the main text because I found this result to be more fundamental than Theorem 1, which focuses on a single $f$.

	+ I also found the full proof of Theorem 2 (in Appendix) difficult to follow. It would be helpful to the reader if additional details are provided. For example, how is the $\lambda_{max}$ related to $\||\psi_{CP}\||\_k$, how does one get $u_{\tilde{\psi}_k,j}$? etc.

2. (Page 7, fourth line) why one should not use RecHalve directly?
3. (Example 1) What is the main difference between the recursion-based reduction proposed in this paper and the recursion-based thinning (KT-SPLIT) of Dwivedi and Mackey?

## Minor Remarks

1. I believe the clarity and presentation of the paper can be significantly improved by fixing the following: grammatical issues, omitted words, defining concepts *before* using them in text. I suggest the paper should be proofread carefully to improve the clarity. I am listing a subset of errors that I found:
	+ (appendix) $ell$ in the text before Lemma 3.
	+ (abstract) The sentence containing "quadratic-time input" did not parse for me.
	+ Some expressions that were used before defining them:  KT ("Our contributions"), 2-thinning  ("Overview of Compress++"), $\sigma^2,c,r_{CP}$ (Page 3, last line)
	+ KT should be KT-SPLIT in last line of Example 1?
	+ space is missing between several words, for example, "simpleyet" (page 6).
	+ $\Gamma_+^{\mathcal F}$ instead of $\Gamma_+^{\mathcal f}$ at several places.
	+ Theorem 4 statement: "is"


2.  Please add citation for the claim of $\Omega(n^{-1/4})$ error in the second paragraph on Page 1.

3. Page 3, Line 1: I suggest changing "closed under multiplication" to "closed under scaling".

4. At several points, the paper uses that $\sigma(\cdot)$ and $c(\cdot)$ are monotonic. It is explicitly mentioned for one calculation (Remark 1)  but not everywhere. Either it should be mentioned everywhere or change the definition to include monotonicity.

5. (Section 3, 1st paragraph) What does "significantly improved error rate" mean?


----
Update: I thank the authors for their response. I am satisfied with the response and continue to recommend accept.


**Summary Of The Paper:**

The topic of the study is distribution-compression/thinning algorithms for Monte-Carlo estimation of functions in RKHS, i.e., given a set of $n$ points, such that uniform distribution on these n points approximates an underlying distribution (with respect to integration over functions in a RKHS) with a certain error, output a set of $m$ points $(m \ll n)$ that has a similar error.
The paper proposes three reductions/meta-algorithms to speed up existing distribution-compression algorithms while maintaining the error guarantee of the original algorithms.
The main result is that the proposed reduction improves the runtime of existing algorithms (with quadratic runtime or more) by a quadratic factor without increasing the corresponding error by a polylog factor.

**Summary Of The Review:**

As mentioned earlier, the paper makes a good contribution and thus I recommend acceptance.

---

> ### Author Response · Authors · 2021-11-22
> **Author Response**
>
> Thank you for the time you’ve taken to review our work and for your positive and constructive feedback!  We are delighted that you found our contributions significant, clear in hindsight, and worthy of acceptance.  We will fix each of the highlighted typos and omissions and carefully proofread the paper to improve clarity.  In addition, we address each of your remaining remarks below.

---

> > ### Author Response · Authors · 2021-11-22
> > **Significance and subtlety of error guarantees**
> >
> > In addition, we will clarify in the revision that while the runtime of compress is immediate upon unrolling the recursion, the error guarantee of Equation 4 is more subtle: here, Compress benefits significantly from random cancellations among the conditionally independent and mean-zero errors from Halve. Without these properties, the errors from each Halve call could compound without cancellation leading to a significant degradation in quality. Fortunately, as we note in Remark 4, we can convert any halving algorithm into one that satisfies the mean zero condition of Thm. 1 without impacting integration error, by symmetrization, i.e., by returning either the outputted half or its complement with equal probability.
> >
> > In the revision we will emphasize that this subtle point can have a substantial impact in practice.  For example, the kernel herding algorithm is completely deterministic and hence does not satisfy the mean zero condition of Thm. 1.  Therefore, our experiments in Figure 3 used a symmetrized version of Herding as the input halving algorithm to recover better-than-iid compression.  However, strikingly (and in line with the theoretical explanation above), if the same experiment is run with standard unsymmetrized kernel herding, the MMD error of Herd-Compress does not decay at all!  Please see the following figure which we will incorporate into the final revision: https://drive.google.com/drive/folders/1Q_hs-jWC2pqMk62Sh00hUHAR5WCPsGu-?usp=sharing

---

> > > ### Comment · Reviewer_FqWv · 2021-11-29
> > > **Thank you for response**
> > >
> > > I thank the authors for their response. I am satisfied with the response and continue to recommend accept.

---

> > ### Author Response · Authors · 2021-11-22
> > **Improved error of Compress++**
> >
> > We will clarify in the revision that the improved error of Compress++ refers to the fact that Compress can inflate input error by a log(n) factor while Compress++ inflates error by at most a constant factor.  We see this improvement in practice as well in the experiments of Sec. 4.

---

> > ### Author Response · Authors · 2021-11-22
> > **RecHalve and the difference between Compress and KT-SPLIT recursions**
> >
> > Recursive halving (RecHalve) is precisely what the KT-SPLIT algorithm uses to compress an input of size n into an output of size n/alpha. Furthermore, the running time of RecHalve is at least as large as the cost of running Halve on an input of size $n$ (so if Halve has quadratic running time, then so does RecHalve).  Thus, the main reason to consider Compress over RecHalve (and main difference between the Compress and KT-SPLIT) is the substantial reduction in running time that Compress offers: Compress reduces quadratic running times to near-linear and reduces the runtime of super-quadratic algorithms by a square-root factor by only running Halve on inputs of size $O(\sqrt{n})$ instead of $n$.

---

> > ### Author Response · Authors · 2021-11-22
> > **MMD guarantee proof sketch and Theorem 2 proof details**
> >
> > Thank you for this suggestion; we will add a proof sketch for the MMD guarantee in the final version.  We also apologize for any confusion and will revise the presentation to make the proof clearer.  In particular, our proof proceeds in several steps. To control the MMD, we need to control the Hilbert norm of the discrepancy of Compress, which we first write as a weighted sum of discrepancies from different (conditionally independent) runs of Halve. To effectively leverage the MMD tail bound assumption for this weighted sum, we reduce the problem to establishing a concentration inequality for the operator norm of an associated matrix. We carry out this plan in four steps summarized below.
> > First, we express the MMD associated with each Halve discrepancy as the Euclidean norm of a suitable vector. Second, we define a matrix dilation operator for a vector that allows us to control vector norms using matrix spectral norms (Lem. 2). Third, we establish moment bounds for these matrices by leveraging tail bounds for the MMD error (Lem. 3). Finally, we prove and apply a subexponential matrix Freedman concentration inequality (Lem. 4--6) to control the MMD error for the Compress output.
> > In the formal argument, the maximum eigenvalue of $M_{k,j}$ is exactly equal to the RKHS norm $||\tilde{\psi_{k,j}}|| $  and  $||\psi_{CP} ||$ equals the maximum eigenvalue of $\sum_{k,j} w_{ \tilde{ \psi_{ k , j } } } M_{k,j} $.

---

### Official Review · Reviewer_aMbB · 2021-11-06

**Correctness:** 4
**Technical Novelty And Significance:** 2
**Empirical Novelty And Significance:** 4
**Recommendation:** 8
**Confidence:** 3

**Main Review:**

The strength of the paper is obvious: it gives a substantial improvement in the running time for a natural and important problem.

The main new idea is very simple. One could even say obvious. In my opinion it's only obvious in hindsight, or obvious to experts in fine-grained complexity (a subfield of theoretical computer science where one tries to classify problems into linear vs quadratic vs cubic, etc.). Therefore, I think this paper is valuable to the broader community.

**Summary Of The Paper:**

This paper gives a meta algorithm for speeding up coreset constructing algorithms for the distribution compression problem.

The benefit of this meta algorithm is that its running time is faster by a square-root factor (e.g. quadratic to linear) while keeping the error rate roughly the same: only a factor of 4 worse.

The method is very simple: split the input into four pieces of size n/4 each, recursively solve each of them, then combine all four answers into a set and take a coreset of this set.

The speed-up in running time is immediate (just because of how recursive formulas work) and the error bounds are also rather clear.

**Summary Of The Review:**

While simple in hindsight, I think the new idea of this paper deserves being published in this venue.

---

> ### Author Response · Authors · 2021-11-22
> **Author Response**
>
> Thank you for the time you’ve taken to review our work and for your positive feedback!  We are delighted that you found our contributions significant, valuable to the broader community, and deserving of being published in this venue.  We strove to make the Compress algorithm as simple and low-overhead as possible to maximize its utility (after developing more complicated meta-algorithms and discarding them in favor of this one), and we agree with the reviewer that the specific Compress algorithm is only obvious in hindsight.

---

> > ### Author Response · Authors · 2021-11-22
> > **Significance and subtlety of error guarantees**
> >
> > In addition, we will clarify in the revision that while the runtime of compress is immediate upon unrolling the recursion, the error guarantee of Equation 4 is more subtle: here, Compress benefits significantly from random cancellations among the conditionally independent and mean-zero errors from Halve. Without these properties, the errors from each Halve call could compound without cancellation leading to a significant degradation in quality. Fortunately, as we note in Remark 4, we can convert any halving algorithm into one that satisfies the mean zero condition of Thm. 1 without impacting integration error, by symmetrization, i.e., by returning either the outputted half or its complement with equal probability.
> >
> > In the revision we will emphasize that this subtle point can have a substantial impact in practice.  For example, the kernel herding algorithm is completely deterministic and hence does not satisfy the mean zero condition of Thm. 1.  Therefore, our experiments in Figure 3 used a symmetrized version of Herding as the input halving algorithm to recover better-than-iid compression.  However, strikingly (and in line with the theoretical explanation above), if the same experiment is run with standard unsymmetrized kernel herding, the MMD error of Herd-Compress does not decay at all!  Please see the following figure which we will incorporate into the final revision: https://drive.google.com/drive/folders/1Q_hs-jWC2pqMk62Sh00hUHAR5WCPsGu-?usp=sharing

---

### Decision · Program_Chairs · 2022-01-20

**Decision:**

Accept (Poster)

**Comment:**

This paper proposes a simple meta algorithm to speed up data thinning algorithms with good theoretical guarantees. The method is both theoretically interesting and useful for practical applications.